# Inter-domain dynamics drive cholesterol transport by NPC1 and NPC1L1 proteins

Piyali Saha[1†], Justin L Shumate[1], Jenna G Caldwell[1], Nadia Elghobashi-Meinhardt[2], Albert Lu[1], Lichao Zhang[3], Niclas E Olsson[4‡], Joshua E Elias[3], Suzanne R Pfeffer[1]*

[1]Department of Biochemistry, Stanford University School of Medicine, Stanford, United States; [2]Department of Chemistry, Technische Universität Berlin, Berlin, Germany; [3]Chan Zuckerberg BioHub, Stanford, United States; [4]Department of Chemical and Systems Biology, Stanford University School of Medicine, Stanford, United States

**Abstract** Transport of LDL-derived cholesterol from lysosomes into the cytoplasm requires NPC1 protein; NPC1L1 mediates uptake of dietary cholesterol. We introduced single disulfide bonds into NPC1 and NPC1L1 to explore the importance of inter-domain dynamics in cholesterol transport. Using a sensitive method to monitor lysosomal cholesterol efflux, we found that NPC1's N-terminal domain need not release from the rest of the protein for efficient cholesterol export. Either introducing single disulfide bonds to constrain lumenal/extracellular domains or shortening a cytoplasmic loop abolishes transport activity by both NPC1 and NPC1L1. The widely prescribed cholesterol uptake inhibitor, ezetimibe, blocks NPC1L1; we show that residues that lie at the interface between NPC1L1's three extracellular domains comprise the drug's binding site. These data support a model in which cholesterol passes through the cores of NPC1/NPC1L1 proteins; concerted movement of various domains is needed for transfer and ezetimibe blocks transport by binding to multiple domains simultaneously.

*For correspondence:
pfeffer@stanford.edu

Present address: †Merck Research Laboratories, South San Francisco, United States; ‡Calico Life Sciences LLC, South San Francisco, United States

## Introduction

NPC1 and NPC1L1 are structurally related, multi-spanning membrane proteins that are important for cholesterol transport in humans. NPC1L1 mediates the uptake of dietary cholesterol at the surface of the intestinal epithelium (*Altmann et al., 2004*; *Weinglass et al., 2008*; *Jia et al., 2011*); the drug, ezetimibe (Zetia) blocks NPC1L1 and is an alternative to statins for patients with elevated plasma cholesterol (*Rosenblum et al., 1998*; *van Heek et al., 2000*; *Davis et al., 2001*). The related NPC1 protein functions in lysosomes to transport LDL-derived cholesterol to the cytoplasm (*Pfeffer, 2019*). Plasma LDL is delivered to the lysosome by endocytosis, and its cholesterol esters are cleaved by acid lipase to release free cholesterol for cellular use (*Brown and Goldstein, 1986*; *Goldstein et al., 1975*). NPC2 protein binds this released cholesterol via its iso-octyl group (*Xu et al., 2007*) and transfers it to the N-terminal domain of NPC1 (*Kwon et al., 2009*). Mutations in either NPC2 or NPC1 protein can give rise to a severe neurodegenerative disorder called Niemann Pick Type C disease, which leads to massive accumulation of cholesterol in lysosomes of all tissues, and premature death (*Pentchev, 2004*).

The availability of the structures of NPC2 (with and without cholesterol; *Friedland et al., 2003* and *Xu et al., 2007*), NPC1 N-terminal domain (*Kwon et al., 2009*), lumenal domains 2 and 3 (hereafter referred to as middle lumenal domain/MLD and C-terminal domain/CTD; *Li et al., 2016a*; *Li et al., 2017b*) and the full length NPC1 by cryoelectron microscopy (*Gong et al., 2016*; *Long et al., 2020*) has catapulted our understanding of these proteins to a new level. Moreover, the co-crystal structure of NPC2 bound to NPC1's second lumenal domain (*Li et al., 2016b*; see also

*Winkler et al., 2019*) provides a valuable starting point for thinking about how cholesterol is likely to be transferred onto NPC1 protein. Despite these important breakthroughs, we still have much to learn about how NPC1 transfers cholesterol across the lysosome membrane after receiving it from NPC2.

It has been proposed that the NPC1 N-terminal domain uses the flexibility of a poly-proline linker to transfer cholesterol to a cavity detected on the other side of the protein, at the so-called sterol-sensing domain (*Kwon et al., 2009*; *Li et al., 2016a*; *Li et al., 2017b*). This cavity lies at the boundary between the inner leaflet of the lysosome membrane and the lumen, an advantageous position as the cavity would be available to both receive cholesterol from NPC1's N-terminal domain and to transfer it to the adjacent membrane. Consistent with this model, *Trinh et al. (2018)* recently found that NPC1's N-terminal domain appears to be able to transfer cholesterol (albeit inefficiently) to an adjacent NPC1 molecule for membrane transfer.

NPC1 domains that receive cholesterol from NPC2 are located ~80 Å from the membrane bilayer (*Gong et al., 2016*). This height corresponds well with the dimensions of the glycocalyx that is thought to line the limiting membrane of the lysosome (*Wilke et al., 2012*). Given the existence of the glycocalyx, it is hard to imagine how NPC1's N-terminal domain could gain access to the lipid bilayer to deliver cholesterol to the membrane. In considering the established mechanisms of transporters for amino acids, sugars, and hydrophobic small molecules, we noted that these extremely diverse proteins undergo significant conformational changes to enable an open extracellular pocket to bind ligand and then close, thereby opening a release site on the opposite side of the protein (and membrane). This type of 'rocker arm' model would require movement of protein domains in relation to one another, for transport functionality. We therefore set out to test whether NPC1 functions by conformational transformations rather than via a sterol hand-off pathway restricted to the N-terminal domain. We present here data consistent with a model in which cholesterol passes directly through NPC1 and NPC1L1 proteins and can be blocked by plugging the channel at the top with a small molecule inhibitor such as ezetimibe for NPC1L1 (this study) or itraconazole for NPC1 (*Long et al., 2020*).

## Results

*Figure 1A* shows the overall domain structure of NPC1 protein, which comprises a cholesterol binding N-terminal domain (red), a second lumenal domain that binds to NPC2 ('MLD', blue; *Deffieu and Pfeffer, 2011*; *Li et al., 2016b*) and a third lumenal domain ('CTD', yellow). The (red) N-terminal domain is attached to the rest of the protein by a potentially flexible polyproline linker sequence (*Figure 1B* at left). Note that the putative sterol-binding site in the sterol-sensing domain (SSD, orange) is located within the membrane bilayer on the side of the protein opposite to the linker domain, adjacent to P691, a residue that is important for NPC1 function (*Watari et al., 1999*; *Ko et al., 2001*; *Ohgami et al., 2004*). To test whether locking the flexible linker in place would inhibit NPC1 function, we relied on the highest resolution (3.3 Å) NPC1 structure (*Li et al., 2017b*) to introduce cysteines that could pair between the top of the polyproline linker and the NPC1 CTD. To design these experiments, we aligned the cryo-EM structure of full-length NPC1 (4.4 Å, PDBID: 3jd8; *Gong et al., 2016*) with the high resolution (3.3 Å) crystal structure of N-terminal domain-deleted NPC1 (PDBID: 5u74 (*Li et al., 2017b*; *Figure 1B*). As shown in *Figure 1C*, introduction of two cysteine residues at positions P251 (in the linker) and L929 (*Figure 1B* inset) did not interfere with the proper folding of NPC1 and its proper delivery to lysosomes as monitored by immunofluorescence microscopy and co-localization with LAMP1.

Perfringolysin O* (PFO*) is a protein that binds cholesterol and can be used to quantitate cholesterol levels in cells lacking NPC1 protein (*Das et al., 2013*; *Li et al., 2015*). The protein is tagged with a fluorescent dye and can be detected by quantitative flow cytometry or light microscopy to monitor lysosomal cholesterol. As shown in *Figure 1D*, NPC1$^{-/-}$ cells lacking transfected NPC1 showed strong PFO* staining; transient expression of wild type NPC1 protein (green) rescued the phenotype and could be easily quantified as a decrease in PFO* signal.

To compare the activities of wild type and mutant proteins more sensitively, we attempted to compare the initial rates of cholesterol clearance by functional or non-functional NPC1 upon expression in NPC1 knockout cells. For this purpose, we synchronized the cholesterol release process in cells expressing rescue constructs by addition of the NPC1-specific inhibitor, U18666A (*Lu et al.,*

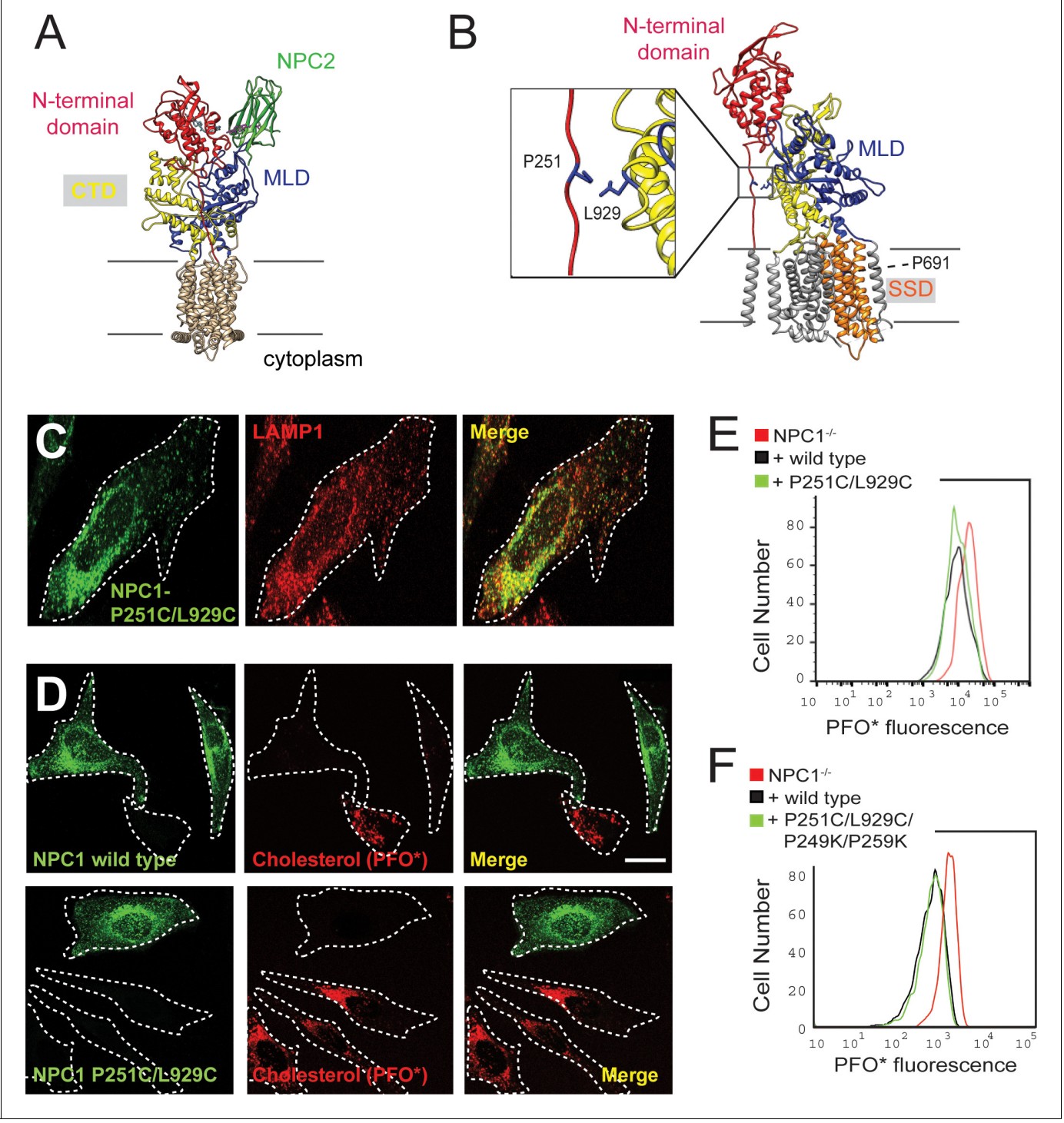

**Figure 1.** Locked N-terminal domain NPC1 rescues cholesterol export from lysosomes. (**A**) Domain structure of NPC1 protein. The N-terminal domain (residues 23–259 including the polyproline linker), middle lumenal domain (MLD, 372–620), and C-terminal domain (CTD, 854–1098) are colored red, blue, and yellow, respectively. (**B**) NPC1 residues mutated to Cys for disulfide bond formation between the polyproline linker and CTD (see inset). The location of the sterol-sensing domain is shown in orange; P691 faces the back. (**C**) Confocal immunofluorescence microscopy of mouse NPC1 P251C/L929C and LAMP1 proteins expressed in HeLa cells (bar, 20 μm). (**D**) Confocal immunofluorescence microscopy of cholesterol accumulation rescue. NPC1$^{-/-}$ HeLa cells were transfected with GFP-mouse NPC1-wild type or P251C/L929C plasmids for 48 h. Thirty-two hours post transfection, cells were incubated with 1 μM U18666a for 16 h; cells were briefly incubated with 10 mM methylamine hydrochloride and chased for cholesterol export for 1 h in 5% LPDS medium, followed by immediate fixation. Intrinsic GFP fluorescence and AF647-PFO* labeling are shown (bar, 20 μm). Images represent maximum intensity projections. (**E**) Flow cytometry of the experiment shown in (**D**). GFP-positive cells of similar intensity were analyzed: NPC1$^{-/-}$, 497

*Figure 1 continued on next page*

*Figure 1 continued*

cells; NPC1 wild type, 478 cells; P251/L929C, 1486 cells. Cell numbers were normalized for comparison. (**F**) Flow cytometry of a rescue experiment using the indicated constructs, carried out as in (**E**). GFP-positive cells were analyzed: NPC1⁻ᐟ⁻, 2968 cells; NPC1 wild type, 2358 cells; P251/L929C/P249K/ P259K, 3906 cells. Flow cytometry analyzed only GFP-positive cells of similar intensity.

The online version of this article includes the following figure supplement(s) for figure 1:

**Figure supplement 1.** Characterization of selected mutant protein glycosylation status or localization.

*2015*) and then monitored the cholesterol export upon removal of the inhibitor. Protein constructs were expressed for ~30 h, followed by overnight treatment of cells with U18666A; we then washed out U18666A and monitored the subsequent relative function of rescue constructs after 1 h, as this short time scale allows for better functional discrimination than is possible when comparing cholesterol accumulation by constructs expressed for 18–24 h. U18666A drug efflux was accelerated by treating cells for 5 min in 10 mM methylamine hydrochloride to raise the pH of endo-lysosomes. The pH was then quickly restored by methylamine washout, and NPC1 function was monitored after 1 subsequent hour in lipoprotein-deficient medium. Control experiments showed that wild type NPC1 required at least 30 min to begin to decrease lysosomal cholesterol by this method.

Using this assay, we found that NPC1 P251C/L929C was as functional as wild type NPC1 in terms of its ability to clear cholesterol from lysosomes, monitored by quantitative flow cytometry or immunofluorescence light microscopy (*Figure 1D,E*). Note that the flow cytometry data have been gated such that only cells with comparable levels of expression of wild type or mutant GFP-tagged proteins are compared.

Critical to the interpretation of these data was a demonstration that a disulfide bridge had formed between residues C251 and C929. We noted that NPC1 P251C/L929C was more efficiently glycosylated in the secretory pathway than either the wild type protein or a single cysteine mutant, consistent with formation of this disulfide bond within the secretory pathway of expressing cells (*Figure 1—figure supplement 1A*). Mature and fully glycosylated NPC1 protein migrates at ~200K upon SDS-PAGE; with endo H treatment to remove high mannose oligosaccharides, a slight mobility decrease is observed (*Figure 1—figure supplement 1*). Less well-folded NPC1 proteins are fully endo H sensitive and yield a band at ~120K upon endo H or protein N-glycanase treatment that may be O-glycosylated (*Schultz et al., 2018*). In our cells, under-glycosylated NPC1 wild type protein was detected, but much less of this form was seen for the P251C/L929C protein. That we detect differences in the steady state glycoforms of NPC1 proteins indicates that they are structurally distinct in cells, consistent with the presence of the additional disulfide bond.

To demonstrate unequivocally the presence of a disulfide bond between NPC1 residues C251 and C929, we used mass spectrometry to detect the disulfide bond in proteolytic digests of the GFP-tagged protein, isolated after expression in cultured cells. Unfortunately, it was difficult to identify the native peptides because of their lengths and polyproline content. To circumvent this challenge, we introduced two lysine residues adjacent to the polyproline stretch in NPC1 P251C/L929C to enhance trypsin cleavage and facilitate disulfide bond detection. *Figure 1F* shows the cholesterol rescue activity of NPC1 P251C/L929C/P249K/P259K. Importantly, this NPC1 mutant was correctly localized to lysosomes (*Figure 1—figure supplement 1B*) and showed full wild type rescue capacity by quantitative perfringolysin O* labeling (*Figure 1F*).

Proteolysis of NPC1 P251C/L929C/P249K/P259K would be predicted to yield disulfide-bonded peptides, with C251 in CQPPPPPMK linked to C929 in NAAECDTY. *Figure 2A* shows the LC-MS elution profiles of both reduced (upper panel) and chemically oxidized (lower panel) forms of synthetic peptides corresponding to these NPC1 peptides (see also *Figure 2—figure supplement 1*). Guided by these peptide standards, we detected the corresponding disulfide species in NPC1 P251C/L929C/P249K/P259K samples expressed in cultured cells followed by protein purification and protease digestion (*Figure 2B* lower panel, C). The disulfide peak was lost upon reduction (*Figure 2B* upper panel), yielding the corresponding thiol peptides. From this analysis we estimate that ~85% of C251 in the NPC1 N-terminal domain is disulfide bonded to C929 in NPC1's C-terminal domain.

To independently quantify the fraction of disulfide-bonded C251 and C929 residues, we labeled any free cysteines in NPC1 P251C/L929C/P249K/P259K with either $^{12}C_2H_2$ ('light') or $^{13}C_2D_2$ ('heavy') iodoacetamide and monitored iodoacetamide labeling of NPC1 protein before and

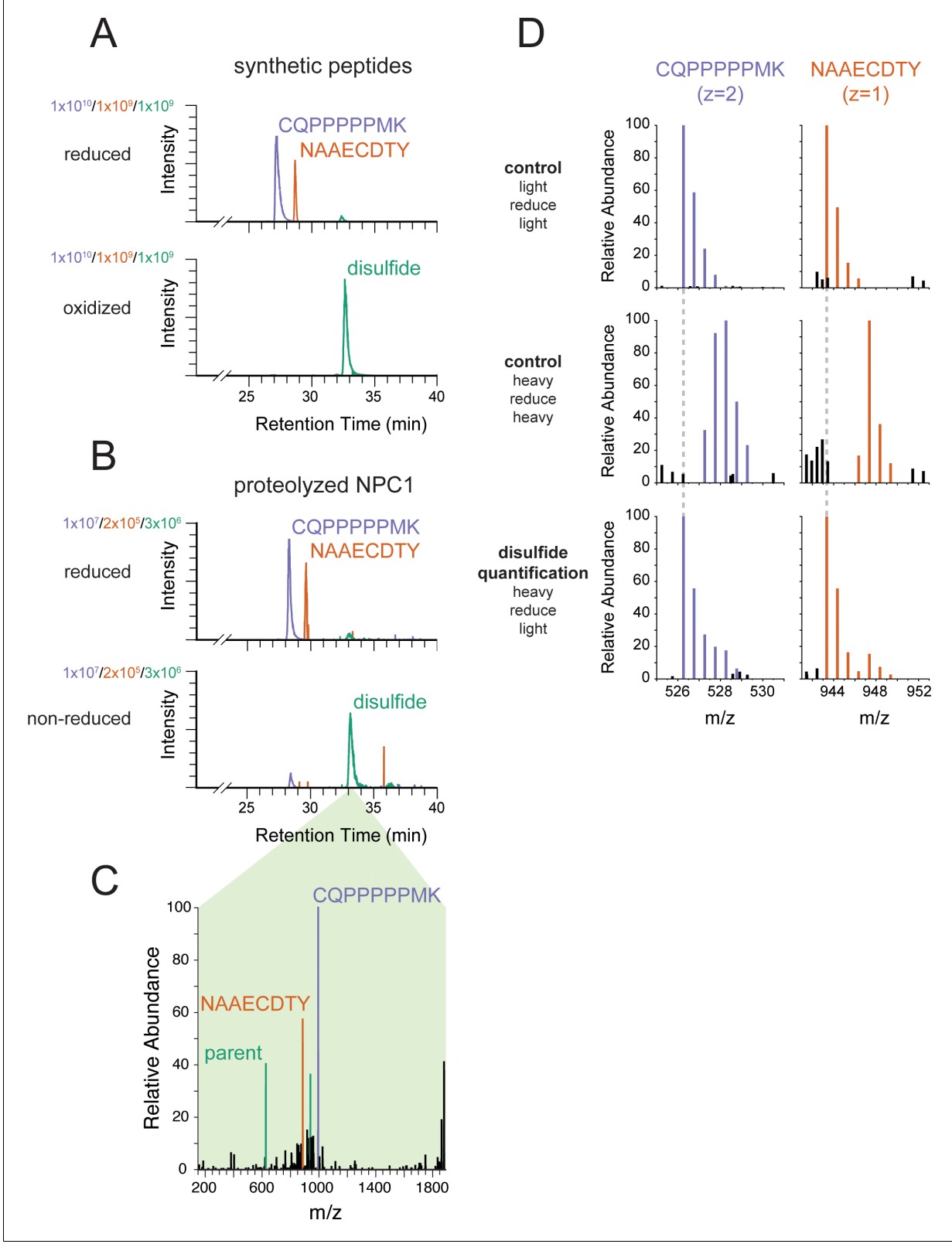

**Figure 2.** Engineered cysteines C251 and C929 form a disulfide bond in NPC1. (**A**) Extracted ion chromatograms from LC-MS analysis of synthetic peptides corresponding to engineered cysteines in NPC1. In both samples, purple traces represent m/z = 526.2569 (corresponding to carbamidomethylated CQPPPPPMK from NPC1 P251C), orange traces represent m/z = 943.3462 (corresponding to carbamidomethylated NAAECDTY from NPC1 L929C), and green traces represent m/z = 626.6005 (corresponding to the disulfide formed between CQPPPPPMK and NAAECDTY). The

*Figure 2 continued on next page*

*Figure 2 continued*

peptide standards CQPPPPPMK and NAAECDTY were produced by solid phase synthesis and either reduced and carbamidomethylated or oxidized to the disulfide using Ellman's reagent. (B) Extracted ion chromatograms from LC-MS analysis of proteolyzed NPC1. Colors as in (A). NPC1 protein was carbamidomethylated in the presence or absence of reducing agent prior to proteolysis. (C) EThcD mass spectrum from NPC1 sample in (B), demonstrating the reductive fragmentation of the putative disulfide precursor into ions with masses corresponding to the two constituent peptides. Peaks matching the mass of the peptide CQPPPPPMK (monoisotopic mass of thiol = 994.49 Da) are colored in purple; peaks matching the masses of the peptides NAAECDTY (monoisotopic mass of thiol radical = 885.32 Da) are colored in orange; peaks matching multiple charge states of the spectrum's parent ions (m/z = 626.60, z = 3, corresponding to the disulfide) are colored in green. (D) MS1 mass spectra of NPC1 peptides whose disulfide content has been quantified using isotope-labeled iodoacetamide. Free thiols in purified NPC1 were labeled with $^{13}C_2D_2$-iodoacetamide ('heavy'). Then the disulfides were reduced and the resulting reactive cysteines were labeled with iodoacetamide lacking isotope labels ('light') followed by proteolysis and LC-MS analysis. Control samples were labeled with the same reagent before and after reduction ('light/reduce/light' and 'heavy/reduce/heavy') to identify isotope distributions in the limiting cases. Colored peaks fall within 10 ppm of expected masses in the isotope envelope of carbamidomethylated CQPPPPPMK (monoisotopic m/z = 526.2569) or carbamidomethylated NAAECDTY (monoisotopic m/z = 943.3462); black peaks correspond to unrelated ions. Dashed gray lines indicate the expected m/z of the monoisotopic peak of the labeled peptides.

The online version of this article includes the following figure supplement(s) for figure 2:

**Figure supplement 1.** HCD mass spectra of disulfide ions (m/z = 626.60) from oxidized disulfide sample in *Figure 2A*.

after chemical reduction. If the protein is fully disulfide-bonded, it should not incorporate $^{13}C_2D_2$ heavy iodoacetamide prior to reduction, enabling us to quantify precisely the fraction of NPC1 protein with a disulfide-protected cysteine. Samples are thus reacted with either light or heavy iodoacetamide, reduced, and then treated again with either light or heavy iodoacetamide. Control experiments using either light/light or heavy/heavy iodoacetamide in both the first and second rounds of labeling (*Figure 2D*, top panels) provided 'standard' spectra for the possible peptide products.

When NPC1 P251C/L929C/P249K/P259K was first reacted with $^{13}C_2D_2$ heavy iodoacetamide, then reduced and subjected to another round of reaction with light iodoacetamide (*Figure 2D*, bottom row), NPC1 C251 and C929 were protected from heavy iodoacetamide labeling prior to chemical reduction—indicating they are disulfide-bonded. The two sites provided consistent measurements of the extent of disulfide bonding: measurement of peptide CQPPPPPMK indicated that 84% of P251C was protected from labeling by participation in a disulfide bond and 88% of the L929C site in the peptide NAAECDTY was protected. This extent of disulfide bonding is consistent with the detected functionality of the NPC1 P251C/L929C/P249K/P259K mutant protein and its correct localization to lysosomes (*Figure 1F*; *Figure 1—figure supplement 1*). All together, these data indicate that movement of the N-terminal domain away from the rest of the protein via the polyproline linker is not required for cholesterol export from lysosomes.

## Inter-domain mobility appears to be important for cholesterol transport

Using photo-reactive, cross-linkable cholesterol, *Hulce et al. (2013)* identified cholesterol-binding peptides proteome-wide; their dataset included NPC1-derived peptides (highlighted in red, *Figure 3A*). The cholesterol-interacting peptides are located at the interface of the MLD and CTD as well as within the cytoplasmic loop connecting transmembrane domain TM7 to TM8. This 14-residue loop composed of residues 800–813 (broken line in *Figure 3A*) was not ordered in the high-resolution crystal structure of N-terminal domain-deleted NPC1 (PDBID: 5u74), and thus is likely mobile. We tested whether this loop is required for NPC1 function by deleting five residues (807-811) in mouse NPC1. The proper folding of this mutant was assessed by monitoring its intracellular localization in NPC1$^{-/-}$ HeLa cells (*Figure 3B*); co-localization with endogenous LAMP1 confirmed that this mutant NPC1 is correctly delivered to lysosomes.

Despite its proper subcellular localization (*Figure 3B*), Δ807–811-mouse NPC1 could not rescue the cholesterol accumulation seen in NPC1$^{-/-}$ HeLa cells, as determined by immuno-fluorescence light microscopy (*Figure 3C*) as well as quantitative flow cytometry of cells expressing comparable amounts of the rescue construct (*Figure 3D*). These residues could contribute mobility for adjacent residues or specific interaction sequences needed for cholesterol export. It is worth noting that this TM7/TM8 loop lies adjacent to a binding site for cross-linkable cholesterol (*Hulce et al., 2013*). We replaced residues 807–811 with alanines, maintaining the length of the loop sequence. Co-localization with endogenous LAMP1 protein confirmed the proper subcellular localization of the mouse

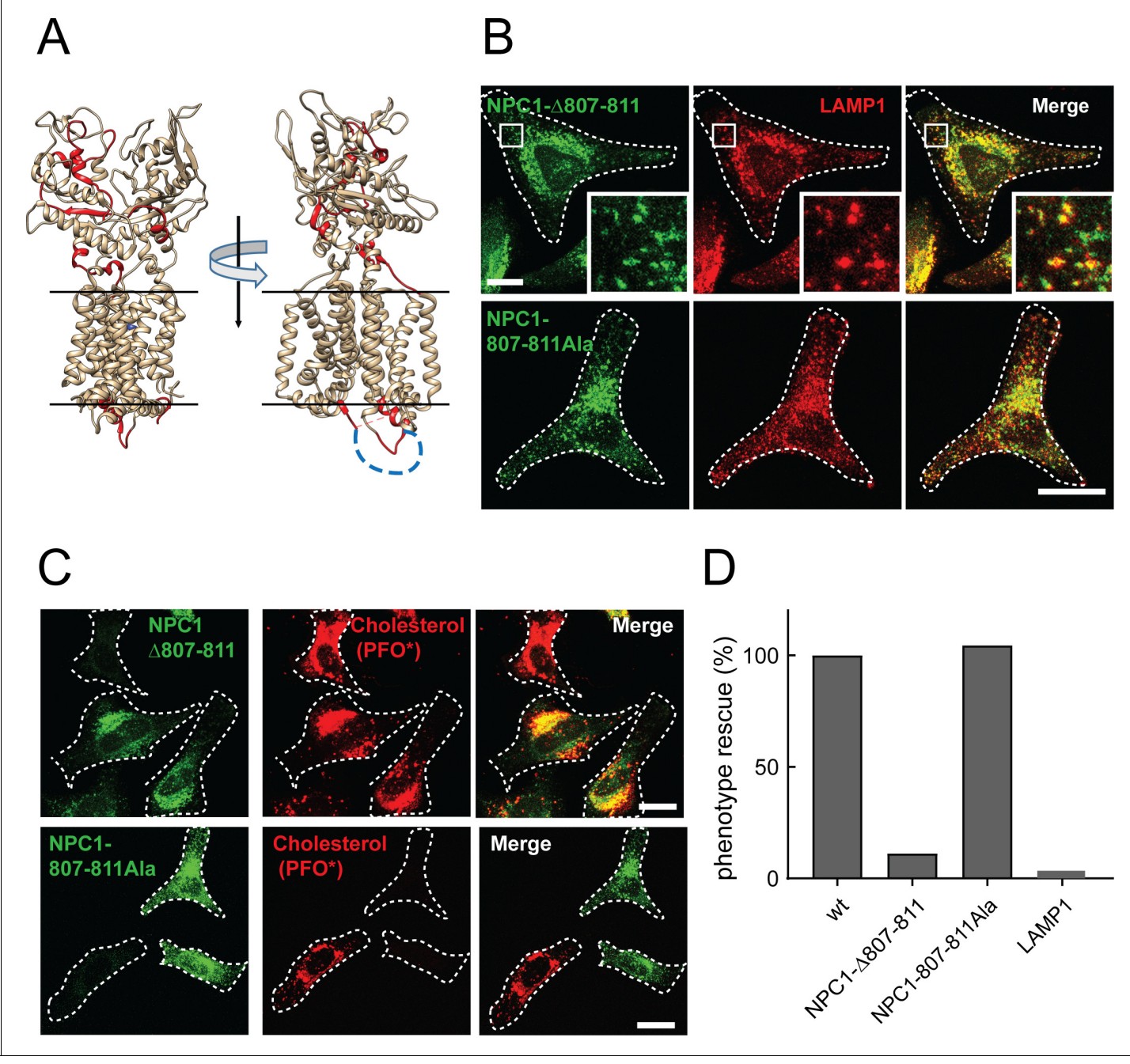

**Figure 3.** NPC1 Δloop mutant cannot rescue cholesterol export from lysosomes. (**A**) Cholesterol-cross-linked peptides (*Hulce et al., 2013*) are highlighted in red for two orientations of the crystal structure of N-terminal domain- and first transmembrane domain-deleted NPC1 (PDBID: 5u74). The disordered cytoplasmic loop residues 800–814 are shown as a blue dotted line. (**B**) Confocal immunofluorescence microscopy analysis of the localization of mouse NPC1-Δ807–811, NPC1-807-811Ala and LAMP1 proteins in HeLa cells (bar, 20 µm). White boxes in images indicate regions of cells enlarged in the insets shown at the lower right of each image. (**C**) Confocal immunofluorescence microscopy of cholesterol accumulation rescue. NPC1$^{-/-}$ HeLa cells were transfected with GFP-tagged mouse NPC1-Δ807–811 or mouse NPC1-807-811Ala plasmids for 48 h and assayed for cholesterol accumulation rescue as in *Figure 1* (bar, 20 µm). (**D**) Quantitation of cholesterol accumulation rescue using flow cytometry. GFP-positive cells with similar expression levels were analyzed: 2480 NPC1; 427 NPC1-Δ807–811; 764 NPC1-807-811Ala; LAMP1 expressing control, 1753 cells counted. Shown are the normalized data from mean fluorescence intensity flow cytometry values.

NPC1-807-811Ala mutant (*Figure 3B*) and similar to wild type NPC1, the 807-811Ala mutant protein fully rescued cholesterol accumulation in lysosomes of NPC1⁻/⁻ transfected cells, monitored by immunofluorescence light microscopy (*Figure 3C*) and quantitative flow cytometry (*Figure 3D*). These data show that the cytoplasmic loop connecting TM7 and TM8 is indispensable for NPC1-mediated cholesterol export, and length is more important than specific amino acid sequences. These data imply that mobility of the protein helices is necessary for transport, with cytoplasmically oriented residues contributing in important ways. It is possible that this small truncation (Δ807–811) may strain the orientation of alpha helices in this region of the protein, but note that these five amino acids were among 14 residues that were not ordered in the high resolution crystal structure (PDBID: 5u74), and the truncation was designed with the goal of maintaining this structure. Nevertheless, the data do not rule out the possibility that this deletion simply favors a conformation of the protein that does not facilitate cholesterol transport.

Cholesterol cross-linked NPC1 peptides also lie across the interface of the MLD and CTD (*Figure 3A*), suggesting that after binding to the N-terminal domain, cholesterol might traverse between interface residues. If so, locking these domains together with a precisely localized disulfide bond should block NPC1 function. To restrict the movement of the MLD and CTD with respect to one another, cysteines were introduced in place of A521 and K1013 to link these domains via a disulfide bond (*Figure 4A*). Co-localization of A521C/K1013C mouse NPC1 with endogenous LAMP1 confirmed the proper subcellular localization of this mutant in lysosomes (*Figure 4B*). The formation of this disulfide bond was also monitored by mass spectrometry; the A521C peptide AP**C**SLNDTSLL was readily detected in reduced samples and recovered at 15% of the level in non-reduced samples, consistent with 85% disulfide bond formation (*Figure 4—figure supplement 1*).

Remarkably, when tested for its ability to rescue cholesterol accumulation, A521C/K1013C-NPC1 failed to rescue cholesterol accumulation in NPC1⁻/⁻ transfected HeLa cells, monitored by immunofluorescence light microscopy (*Figure 4C*) and quantitative flow cytometry (*Figure 4D*). As an additional control, we also monitored the activity of an NPC1 protein in which only one cysteine mutation was introduced: NPC1⁻/⁻ HeLa cells transfected with A521C-mouse NPC1 were fully rescued for cholesterol accumulation, monitored by light microscopy and flow cytometry (*Figure 4C,D*). Together, these data suggest that NPC1 only functions when the MLD and CTD can move in relation to one another. This essential mobility is consistent with a model in which cholesterol passes through the interface between the MLD and CTD as part of the cholesterol transport process.

## Molecular dynamics simulations of NPC1 and mutant proteins

We sought additional hints to the mechanism of NPC1 cholesterol transport using molecular dynamics (MD) simulations. Conformational dynamics of wild type and the above-mentioned NPC1 protein constructs were analyzed by measuring the RMSD (root mean square deviation in Å) of protein backbone atoms, as sampled during the MD trajectories (*Figure 5*). NPC1 dynamics were characterized by a high degree of flexibility of the linker region (residues 247–266) connecting the N-terminal domain and helix 1 of the transmembrane domain, as well as a high degree of disordered secondary structure in the loops facing the cytoplasm (see *Figure 1A*). Notably, during our simulations, the N-terminal domain maintained its interface with the MLD and CTD and did not exhibit any large-scale hinging motion toward the transmembrane domains. The NPC1 mutant in which the MLD and CTD are locked together (A521C/K1013C) showed a similar range of motion for protein backbone atoms to that of wild type NPC1 (*Figure 5*). On the other hand, mutants in which the N-terminal domain was locked to the CTD (P251C/L929C) showed smaller RMS deviations relative to wild type. Locking the relatively disordered linker region of the N-terminal domain to the CTD removed a large source of protein flexibility. Similarly, the large RMSD values measured with mutant Δ807–811 stem primarily from increased disorder in the loops on the cytoplasmic side of the membrane (*Figure 5*).

To evaluate the extent of long-range concerted motion between protein domains, the distance correlation coefficients (DiCC) were calculated between the four domains (NTD, MLD, CTD, and the transmembrane domains; *Table 1*). For highly correlated motion between protein domains, the DiCC approach 1.00, whereas for uncorrelated motion the DiCC approach zero. In the wild type protein, MLD and the TMD showed the highest degree of correlated motion, whereas all mutants exhibit altered behavior. P251C/L929C showed the highest correlation between the transmembrane domains and CTD. In A521C/K1013C, the largest DiCC was calculated between the MLD and CTD

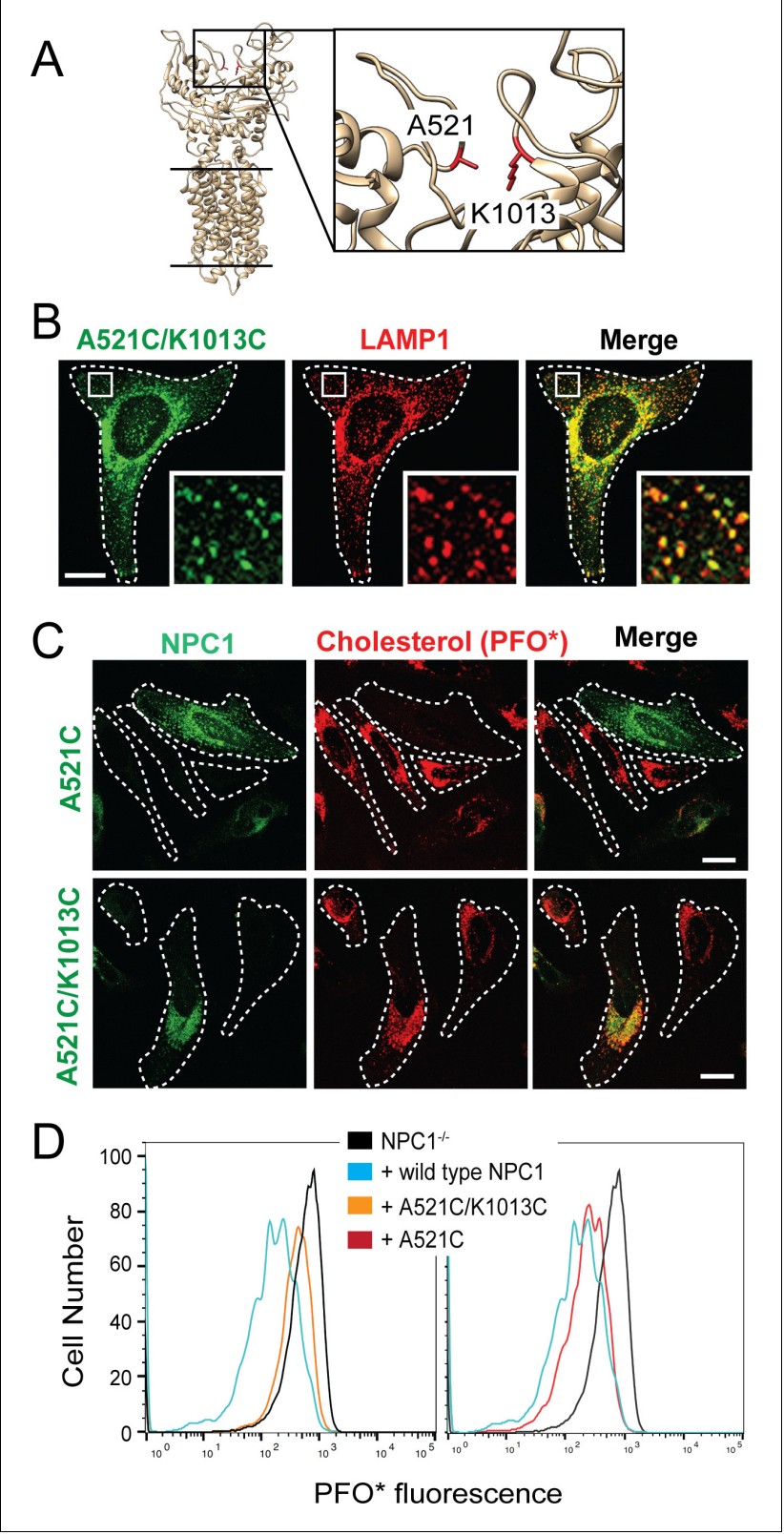

**Figure 4.** NPC1 Disulfide bond-locked MLD and CTD fails to rescue cholesterol export from lysosomes. (**A**) Partial NPC1 structure; inset, close-up view of the MLD/CTD interface. The amino acid residues mutated to cysteines for disulfide bond formation are shown and highlighted in red. (**B**) Confocal immunofluorescence microscopy analysis of mouse NPC1-A521C/K1013C and LAMP1 proteins in HeLa cells (bar, 20 μm). White boxes in images indicate
*Figure 4 continued on next page*

*Figure 4 continued*

regions of cells enlarged in the insets shown at the lower right of each image. (C) Confocal immunofluorescence microscopy of cholesterol accumulation rescue for NPC1-A521C or mouse NPC1-A521C/K1013C. (D) Flow cytometry of the rescue experiment analyzed in (C). GFP-positive cells with similar expression levels were analyzed: 17746 NPC1[-/-] cells; 1315 NPC1 wild type; 1137 NPC1-A521C/K1013C cells; 837 NPC1-A521C cells; cell numbers were normalized for comparison.

The online version of this article includes the following figure supplement(s) for figure 4:

**Figure supplement 1.** Extracted ion chromatograms from LC-MS analysis of proteolyzed A521C/K1013C NPC1.

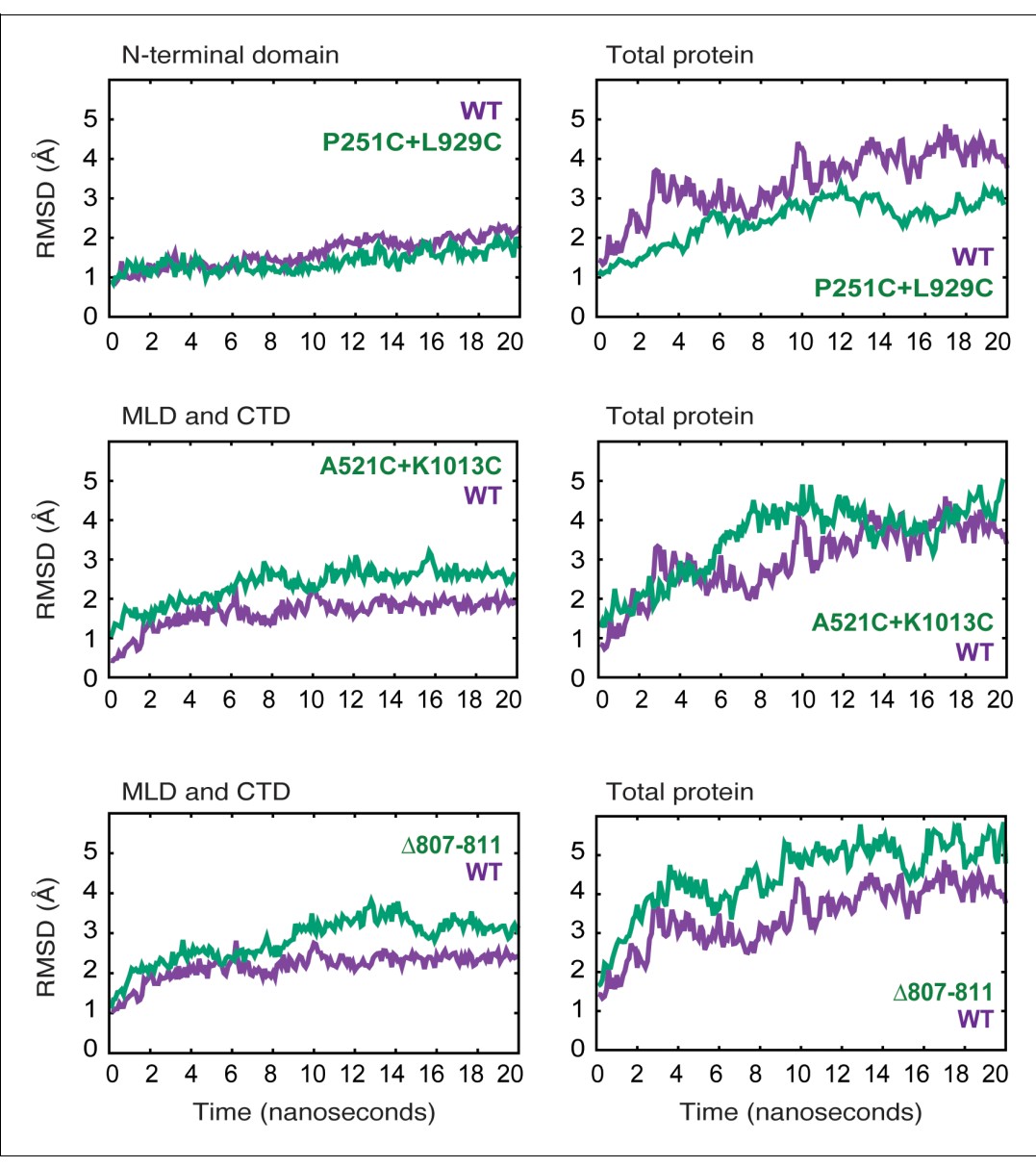

**Figure 5.** Molecular dynamics simulations of NPC1 wild type and mutant proteins. RMSD (Å) of protein backbone atoms for each simulated model is plotted as a function of time for the indicated mutants in relation to their wild type counterparts.

**Table 1.** Distance correlation coefficients for mutants analyzed.
Strongest inter-domain correlations are indicated in bold.

**A521C+K1013C**

|       | NTD   | MLD   | CTD   | TMD   |
|-------|-------|-------|-------|-------|
| NTD   | –     | –     | –     | –     |
| MLD   | –     | 1.000 | **0.850** | 0.810 |
| CTD   | –     | **0.850** | 1.000 | 0.594 |
| TMD   | –     | 0.810 | 0.594 | 1.000 |

**P251C+L929C**

|       | NTD   | MLD   | CTD   | TMD   |
|-------|-------|-------|-------|-------|
| NTD   | 1.000 | **0.969** | 0.945 | 0.931 |
| MLD   | **0.969** | 1.000 | 0.919 | 0.942 |
| CTD   | 0.945 | 0.919 | 1.000 | 0.952 |
| TMD   | 0.931 | 0.942 | 0.952 | 1.000 |

**Δ807-811**

|       | NTD   | MLD   | CTD   | TMD   |
|-------|-------|-------|-------|-------|
| NTD   | 1.000 | 0.496 | 0.522 | 0.391 |
| MLD   | 0.496 | 1.000 | **0.836** | 0.398 |
| CTD   | 0.522 | **0.836** | 1.000 | 0.395 |
| TMD   | 0.391 | 0.398 | 0.395 | 1.000 |

**WT**

|       | NTD   | MLD   | CTD   | TMD   |
|-------|-------|-------|-------|-------|
| NTD   | 1.000 | 0.569 | 0434  | 0.529 |
| MLD   | 0.569 | 1.000 | 0.751 | **0.811** |
| CTD   | 0.434 | 0.751 | 1.000 | 0.617 |
| TMD   | 0.529 | **0.811** | 0.617 | 1.000 |

(0.850), reflecting the concerted domain motion as a result of the disulfide bridge locking these domains together.

Taken together, these results highlight the extent to which local changes in the NPC1 protein are propagated through the entire protein, affecting long-range domain motion. Moreover, the data support a model in which NPC1 protein relies on 'cross-talk' between domains, and these mechanisms are likely employed during sterol transfer.

## Inter-domain mobility is also important for cholesterol uptake by NPC1L1

Because NPC1L1 mediates cholesterol transport from the cell surface, we could test whether inter-domain flexibility is also important for this related cholesterol transporter using orthogonal assays. *Figure 6A* shows a model of NPC1L1 obtained by threading its sequence onto the structure of NPC1 using Swiss-Model (PDB: 5u74; *Waterhouse et al., 2018*). NPC1L1 mutants that were constrained at their cytoplasmic loop (Δ820–824) or restricted in terms of the mobility of the MLD in relation to the CTD (F532C/I1022C) were designed in a manner analogous to the NPC1 mutants; these failed to import cholesterol into HEK293T cells expressing these proteins at the cell surface (*Figure 6B*). As shown previously (*Zhang et al., 2011*), NPC1L1 missing its N-terminal domain also failed to import cholesterol efficiently, although a low level of uptake was observed when compared with control samples (*Figure 6B*). Lack of transport was not a result of differences in cell surface localization or protein levels, as we determined the localizations and amounts of all the mutant proteins using a cell surface biotinylation assay in conjunction with immunoblotting (*Figure 6C* and *Johnson and Pfeffer, 2016*). These experiments confirm the importance of interdomain flexibility for cholesterol transport by both NPC1 and NPC1L1 proteins.

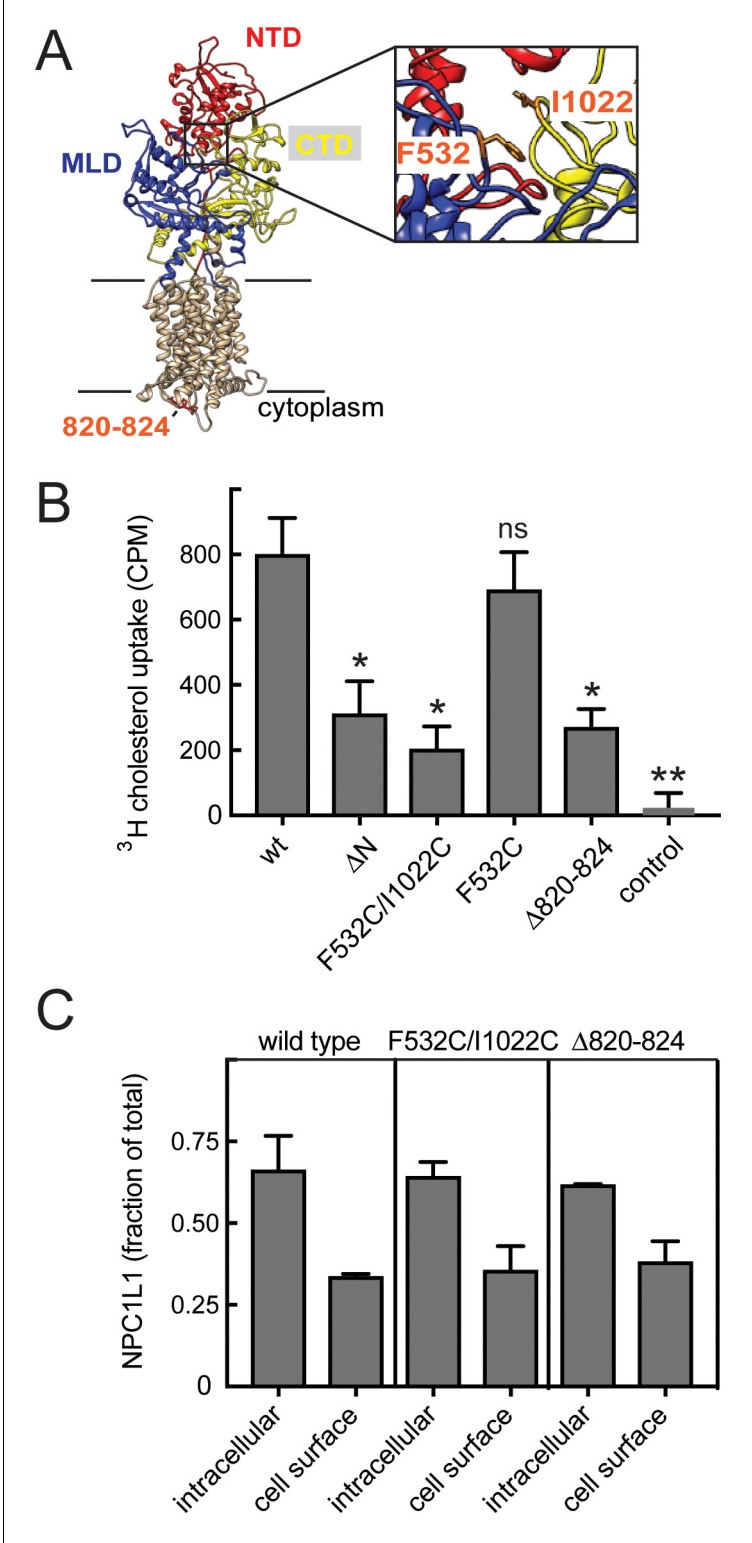

**Figure 6.** Inter-domain mobility is required for cholesterol transport by NPC1L1 protein. (**A**) Structure model of NPC1L1 built by Swiss-Model using NPC1 (PDBID: 5u74) as template for UNIPROT Q9UHC9-1. Extracellular domains are labeled and colored as in *Figure 1*; NTD, residues 33–275; MLD, 373–632; CTD, 861–1111. In orange are residues 820–824, F532 and I1022 (enlarged in a slightly rotated inset at right). (**B**) Cholesterol uptake by cells expressing the indicated constructs. p values determined by unpaired t test were ΔN, 0.03; F523/I1022C, 0.01; F532C, ns; Δ820–824, 0.01; control, 0.004. (**C**) Localization of NPC1L1 proteins determined by cell surface

*Figure 6 continued on next page*

*Figure 6 continued*

biotinylation. HEK293T cells transfected with GFP-tagged constructs were surface-labeled with Sulfo-NHS-biotin-EZ-link on ice for 30 min and biotin-labeled proteins captured on streptavidin beads and analyzed by immunoblot using anti-GFP antibody. Quantitation of the LI-COR data is shown; error bars represent SEM of triplicate determinations. Mutant values were statistically indistinguishable from wild type (all p values not significant).

## The MLD plays a unique role in NPC1L1

We have shown previously that NPC1's MLD binds NPC2 to facilitate transfer of cholesterol from NPC2 onto NPC1's N-terminal domain (*Deffieu and Pfeffer, 2011*; *Li et al., 2016b*). Unlike NPC1, NPC1L1 does not receive cholesterol from an NPC2-like protein in the intestine—instead, it receives cholesterol from bile salt micelles (*Yamanashi et al., 2007*; *Haikal et al., 2008*). We were intrigued by the possibility that the MLD of NPC1L1 might play an analogous role to its related domain in NPC1 and that it might bind bile salt micelles to facilitate transfer of cholesterol onto NPC1L1's N-terminal domain.

To test this, we established a binding assay using purified, soluble NPC1L1 N-terminal domain or NPC1L1 MLD. [3]H-cholesterol was presented to these proteins either mixed with Nonidet P40 (NP-40) below its critical micelle concentration (CMC), according to established methods used to measure cholesterol binding to NPC1 and NPC1L1 NTD (*Infante et al., 2008*; *Kwon et al., 2011*), or in the form of a bile salt micelle (*Figure 7*; *Figure 7—figure supplement 1*).

As expected, purified NPC1L1 N-terminal domain bound cholesterol when it was presented in the presence of sub-CMC NP-40 (*Kwon et al., 2011*; *Figure 7* top); under these conditions, no cholesterol binding was detected to purified NPC1L1 MLD. In contrast, when cholesterol was presented to the N-terminal domain in the form of a bile salt micelle, no binding was detected, but significant binding was observed for reactions containing NPC1L1 MLD (*Figure 7*, bottom). These data indicate that NPC1L1 MLD contains a specific binding site for cholesterol in the form of a bile salt micelle.

Binding of cholesterol in the form of a bile salt micelle to the MLD was not altered by addition of up to 100 µM of the NPC1L1 inhibitor, ezetimibe, which relies on NPC1L1 MLD residues for high-affinity binding (*Weinglass et al., 2008*). These experiments reveal, for the first time, that MLD may function to bind bile salt micelles and thus facilitate capture of cholesterol by the N-terminal domain as part of the cholesterol uptake process. In addition, they suggest that ezetimibe inhibition of NPC1L1 protein is likely not a result of interference with bile salt micelle binding to MLD.

## Domain interface-locking contributes to the mechanism of ezetimibe inhibition

The mechanisms by which small molecules inhibit transporter proteins can provide important clues to the mechanisms by which transporters normally function. As mentioned earlier, sequences located within NPC1L1 MLD are important for high-affinity ezetimibe binding (*Weinglass et al., 2008*). We determined the possible additional contributions of other NPC1L1 domains to ezetimibe binding. As shown in *Figure 8A*, cells expressing wild type NPC1L1 showed binding to [3]H-ezetimibe that was specific as it could be competed away by addition of excess unlabeled ezetimibe. Expression of an N-terminal domain-deleted NPC1L1 protein abolished ezetimibe binding, despite the efficient delivery of ΔN-NPC1L1 to the plasma membrane of transfected HEK293T cells; ~30% of both wild type and mutant NPC1L1 proteins were localized to the cell surface (*Figure 8B*). Thus, NPC1L1 N-terminal domain sequences are also important for ezetimibe binding.

It is important to note that the MLD residues previously identified as being important for ezetimibe binding by *Weinglass et al. (2008)* are predicted to lie at the interface between the MLD and CTD in the structure model. Given the apparent importance of inter-domain interfaces in cholesterol transport by NPC1, we tested the possibility that inter-domain interface residues create a binding site for ezetimibe on NPC1L1. Small molecule binding adjacent to interface residues would have the same consequence as the disulfide-bonded, locked mutants—it could restrain protein dynamics and therefore block cholesterol transport.

*Figure 9A* shows the effect of mutating a number of residues that are predicted to be important for contacts between NPC1L1 N-terminal domain and the MLD or CTD. *Table 2* summarizes the rationale for each of the mutations generated. Remarkably, single point mutations of representative

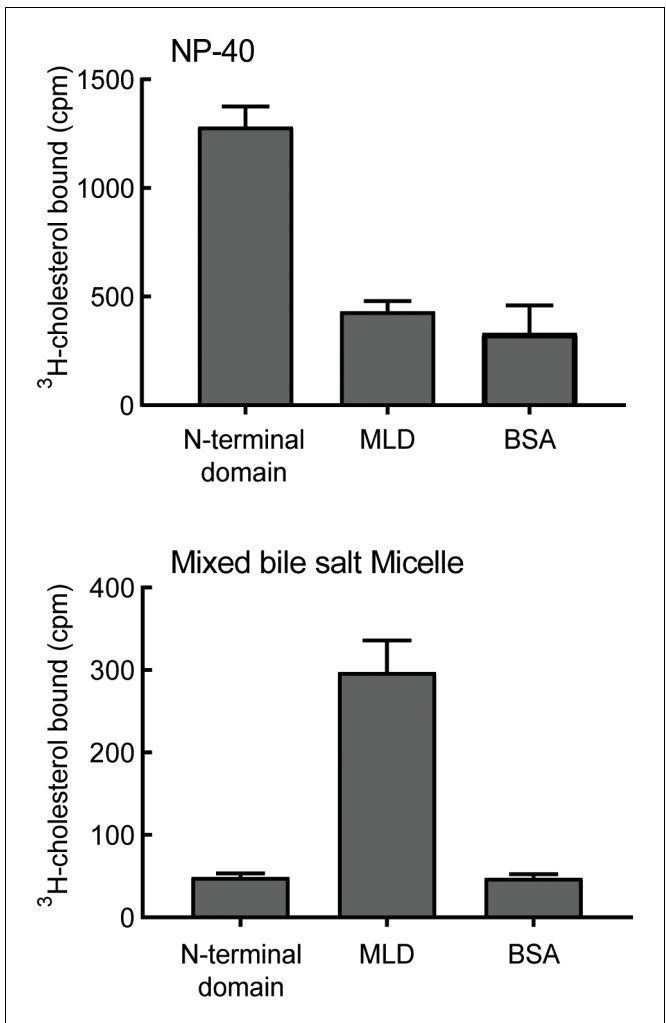

**Figure 7.** NPC1L1 MLD binds bile salt micelles. Binding of purified NPC1L1 N-terminal domain or MLD to [3]H-cholesterol, delivered either in sub-CMC NP-40 (**A**) or in mixed bile salt micelles (**B**). Error bars represent SEM for triplicate determinations from a representative experiment from five independent experiments.

The online version of this article includes the following figure supplement(s) for figure 7:

**Figure supplement 1.** Purified NPC1L1 N-terminal domain and NPC1L1 MLD used in *Figure 6*.

---

residues from each of these interfaces abolished the ability of ezetimibe to bind NPC1L1 (*Figure 9A*). Once again, we verified the cell surface localization of many of these mutant proteins; in each case, about 30% of the protein was present at the cell surface (*Figure 9B*). These data provide new information about the structural basis for ezetimibe interaction with NPC1L1 protein and support a model whereby ezetimibe blocks NPC1L1 by locking the various domains together, potentially plugging a potential cholesterol channel through the molecule (*Winkler et al., 2019*; *Long et al., 2020*).

## Discussion

We have shown here that dynamic, inter-domain interactions within NPC1 and NPC1L1 are essential for the ability of these proteins to mediate cholesterol transport out of lysosomes or across the plasma membrane, respectively. Specifically, mutants that can lock the MLD to the CTD at the interface of these domains interfere with cholesterol transport; residues that comprise a loop at the cytoplasmic face of the transmembrane domains are needed to provide inter-domain mobility and/or an

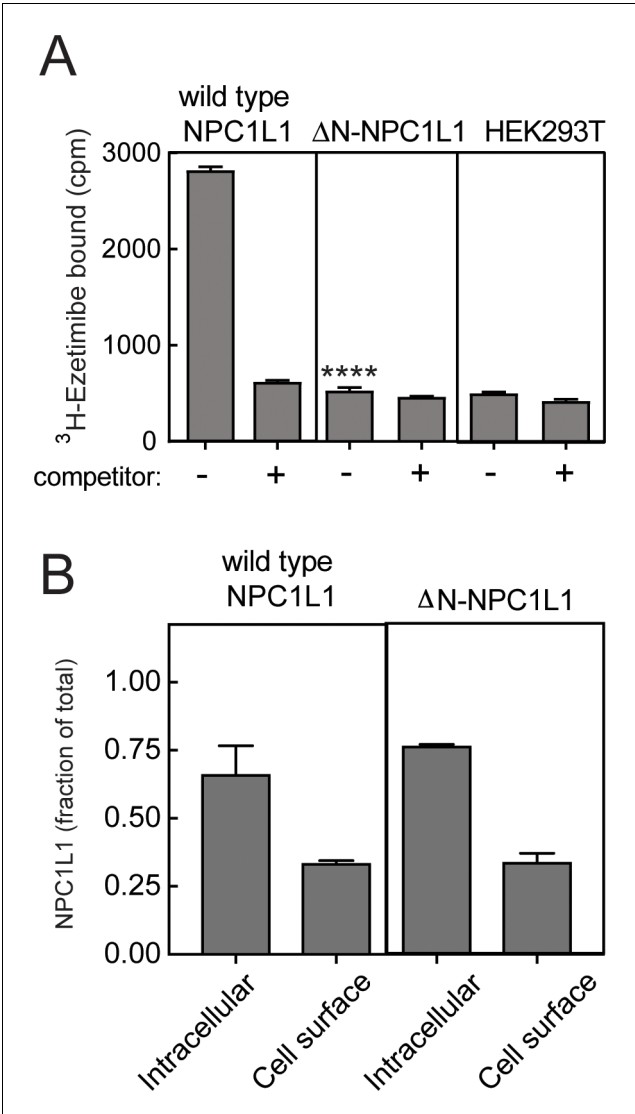

**Figure 8.** NPC1L1 N-terminal domain is essential for ezetimibe binding at the cell surface. (A) $^3$H-ezetimibe binding to HEK293T cells transfected with either wild type or N-terminal domain deleted ΔN-NPC1L1. Forty-eight hours post transfection, cells were incubated with 50 nM $^3$H-ezetimibe for 2 h at 37°C ± 10 μM cold ezetimibe. Untransfected HEK293T cells were used as control. Equal protein amounts of total cell lysates were analyzed by scintillation counter. Error bars represent SEM for a representative experiment carried out in triplicate. The p value for ΔN-NPC1L1 compared with wild type was <0.001 without competitor. (B) Localization of NPC1L1 proteins determined by cell surface biotinylation as in *Figure 5*; error bars represent SEM of triplicate determinations. There was no significant difference between wild type and mutant samples as determined by unpaired t test.

unconstrained conformation, and can be replaced with polyalanine for full function. Moreover, we have shown that ezetimibe inhibition of NPC1L1 relies on its interaction with multiple domains and depends on sequences that are predicted to contribute to domain interface interactions in the protein structure. A highly plausible model for ezetimibe inhibition would be that it functions in a manner analogous to the disulfide bond mutants: locking the domains to one another and thereby blocking cholesterol transport. By binding at the interface, ezetimibe could also plug a channel between these domains. We showed for the first time that purified NPC1L1 MLD binds bile salt micelles. Although we cannot exclude the possibility that ezetimibe blocks bile salt interaction in full-length NPC1L1, the mutagenesis experiments support the importance of interface residues for drug binding, which will have the additional consequence of blocking inter-domain dynamics.

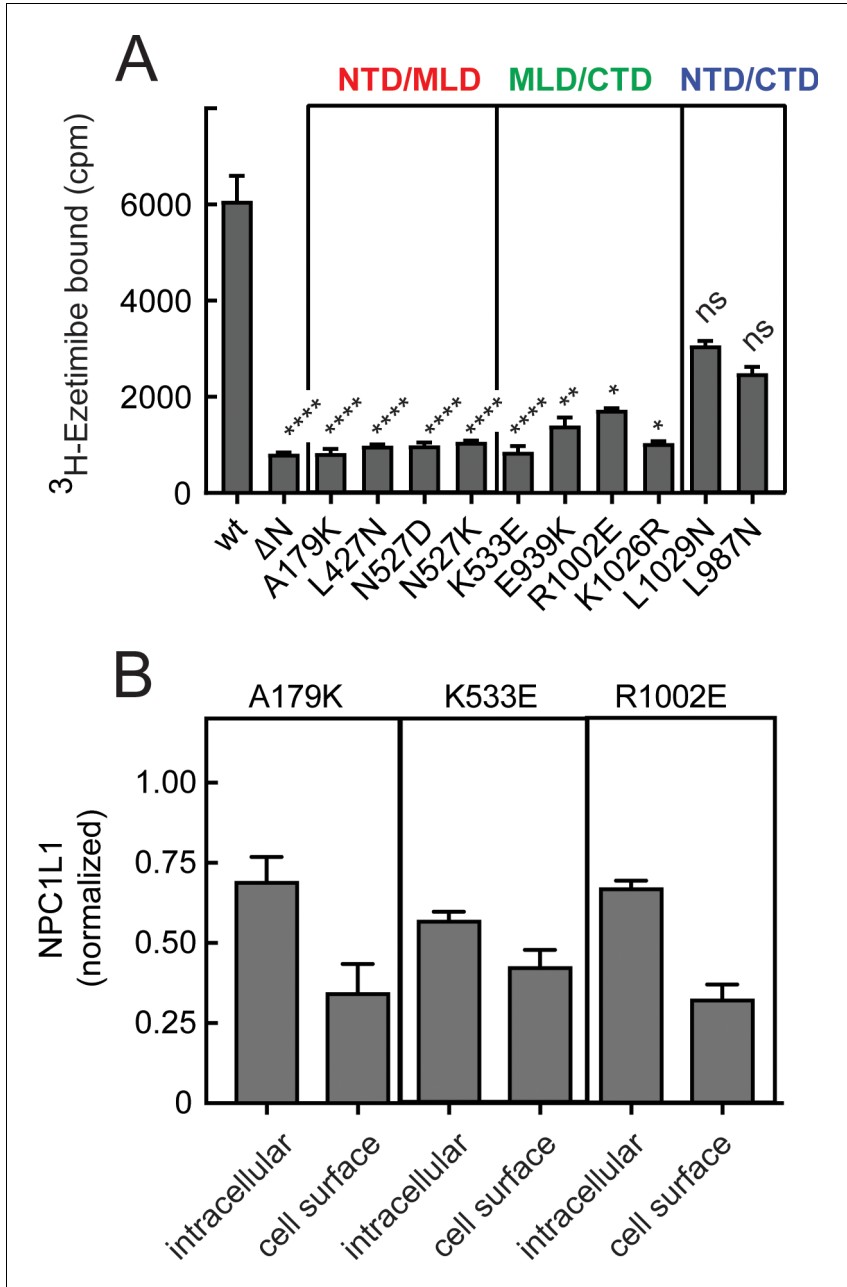

**Figure 9.** Inter-domain interfaces of NPC1L1 are critical for ezetimibe binding. (**A**) $^3$H-ezetimibe binding of HEK293T cells transfected with either wild type NPC1L1 or mutant NPC1L1 proteins designed to disrupt the interfacial interactions between domains of NPC1L1. The colored labels in (**A**) represent interfaces altered by the mutants shown. Error bars represent SEM for triplicate samples. p values compared with wild type were <0.0001 for all mutants except E939K, 0.004; R1002E, 0.03; K1026R, 0.011; L1029N, ns; L987N, ns. (**B**) Localization of representative NPC1L1 mutants as in *Figure 5*; LI-COR data quantitation is shown and error bars represent SEM for triplicate samples. Intracellular and cell surface values were not significantly different for K533W and R1002E samples compared with A179K determined by unpaired t test.

It is currently accepted that NPC2 transfers bound cholesterol to the NPC1 N-terminal domain; however, the subsequent steps by which cholesterol transporters move cholesterol across the membrane remain unclear. Determination of the structure of NPC1 revealed that the N-terminal domain cholesterol binding site is located on the opposite side of the protein to the sterol-sensing domain; this finding led to the proposal that the N-terminal domain may transfer its cholesterol to an

**Table 2.** Rationale for NPC1L1 domain interface mutants generated.

| Residue | Residues in other domains within 5 Å | Type of interaction | Mutation chosen |
|---|---|---|---|
| A179 | L523, Y524, N527 | Hydrophobic/hydrophilic | K (possibly trigger steric clash and disrupt the interface) |
| L427 | A180 | van der waals/hydrophobic | N (possibly disrupt the hydrophobic interface) |
| N527 | R112, Q257 | Hydrophilic | D (could stabilize the interface, restricting domain flexibility); K (potentially destabilize the interface because of positive charge repulsion) |
| K533 | K1026 | Electrostatic | E (could stabilize and restrict domain flexibility) |
| E939 | P549, F551 | van der waals | K (longer sidechain might result in steric clash, disrupt the interface) |
| R1002 | D250 | Electrostatic | E (possibly disrupt the interface) |
| K1026 | K533 | Electrostatic | R (possibly stabilize the interface and restrict domain flexibility) |
| L1029 | L530, M543 | van der waals/hydrophobic | N (possibly disrupt the interface) |
| L987 | I72 | van der waals/hydrophobic | N (possibly disrupt the interface) |

adjacent NPC1 molecule (*Li et al., 2016a*). The experiments presented here do not address this possibility directly, but they demonstrate that cholesterol transport does not require full release of the N-terminal domain from the rest of the protein. Locking the polyproline linker that attaches the N-terminal domain to the rest of NPC1 protein had no impact on the ability of NPC1 to export cholesterol from lysosomes using kinetically sensitive and quantitative assays for this process. These results suggest that the N-terminal domain need not hinge away from the rest of the protein to transfer cholesterol from the N-terminal domain to other parts of the protein and/or across the membrane. Note that the portion of the linker most proximal to the NTD will nevertheless retain some flexibility, despite the presence of the additional disulfide bond (*Figure 1B*).

Taken together, our data support a model in which cholesterol can pass directly through NPC1, either to the sterol-sensing domain or all the way through the membrane bilayer. This model is further supported by recent computations and structural experiments predicting a cholesterol transfer path through the center of NPC1 (*Elghobashi-Meinhardt, 2019*; see also *Wheeler et al., 2019*; *Winkler et al., 2019*; *Long et al., 2020*). Because of the importance of cytoplasmically oriented residues for NPC1 function, we favor a model in which the sterol-sensing domain binding site represents a regulatory site and is not necessarily involved in the transport process per se.

Precedent for cholesterol passing through a related transporter comes from the structural work of Nieng Yan and Xiaochun Li and their colleagues who have recently reported structures of the related Patched (PTCH) protein, which is important for Hedgehog signaling (*Gong et al., 2018*; *Qi et al., 2018a*; *Qi et al., 2018b*). PTCH has 12 transmembrane domains instead of 13; it also lacks the cholesterol binding N-terminal domain of NPC1 and NPC1L1. Transmembrane domains 2–6 comprise a sterol-sensing domain analogous to transmembrane domains 3–7 of NPC1. Yan and colleagues detected two sterol molecules in the PTCH structure: one in a cavity located between the two extracellular domains and the second, adjacent to the sterol-sensing domain (*Gong et al., 2018*). Their mutant analysis suggests that cholesterol binding causes significant conformational changes, with an untwisting of the interactions between the extracellular domains and reorientation of certain TMs. *Qi et al. (2018b)* found two sterol-like ligand densities in the transmembrane domain

of each PTCH molecule, one in the sterol-sensing domain and the other near transmembrane domain 12. Importantly, the structural analysis is consistent with a tunnel in the PTCH protein, through which a sterol could pass.

Does NPC1 also bind cholesterol in a cavity between its extracellular domains, analogous to PTCH? Does NPC1 contain a similar channel through which cholesterol might pass? Important hints came initially from Cravatt and colleagues (*Hulce et al., 2013*), whose proteome-wide analysis of cholesterol binding proteins revealed novel cholesterol binding locations in NPC1 (*Figure 3A*). Our mutant protein functional analysis is consistent with the possibility that NPC1 and NPC1L1 undergo significant conformational transitions to accomplish transport of cholesterol across the membrane.

While this study was under review, Xiaochun Li's team reported the cryo-EM structure of NPC1 with an added inhibitor, itraconazole, occupying a central cleft at the top of NPC1's MLD and CTD (*Long et al., 2020*). This finding meshes well with our proposal for how ezetimibe blocks cholesterol transport by NPC1L1 at a similar location in the corresponding structure. Whether cholesterol exits this channel at the sterol-sensing domain (see also *Winkler et al., 2019*) or proceeds through the molecule to the other side of the bilayer is not yet clear, but our finding that constriction of the proteins' cytoplasmic domains blocks function suggests that the channel may indeed pass all the way through the transmembrane domain. Structural analysis of such mutant proteins may help to resolve this question.

# Materials and methods

## Key resources table

| Reagent type (species) or resource | Designation | Source or reference | Identifiers | Additional information |
|---|---|---|---|---|
| Gene *Mus musculus* | NPC1 | PMID:27551080 | | |
| Gene (*Homo sapiens*) | NPC1L1 | PMID:27075173 | | |
| Gene (*Homo sapiens*) | LAMP1 | PMID:27664420 | | |
| Cell line (*Homo sapiens*) | HeLa NPC1 KO | PMID:26578804 | | |
| Cell line (*Homo sapiens*) | HEK293T | ATCC | | |
| Cell line (*Homo sapiens*) | 293F | ATCC | | |
| Cell line (*Spodoptera frugiperda*) | SF9 | ATCC | | |
| Transfected construct *Mus musculus* | NPC1-eGFP | PMID:27551080 | | |
| Transfected construct (*Homo sapiens*) | NPC1L1-eGFP | PMID:27075173 | | |
| Transfected construct (*Homo sapiens*) | LAMP1-eGFP | PMID:27664420 | | |
| Transfected construct (*Homo sapiens*) | pFastBac-NPC1L1-wt-N-terminal domain | PMID:27075173 | | |
| Transfected construct (*Homo sapiens*) | pCMV-FLAG-His6-NPC1L1-MLD | PMID:27075173 | | |

*Continued on next page*

*Continued*

| Reagent type (species) or resource | Designation | Source or reference | Identifiers | Additional information |
|---|---|---|---|---|
| Antibody | Chicken polyclonal anti-GFP | Life technologies | | 1:1000 |
| Antibody | IRDye 680RD polyclonal Donkey anti-chicken | LI-COR | 926–68075 | 1:10000 |
| Antibody | IRDye 800CW streptavidin | LI-COR | 926–32230 | 1:10000 |
| Antibody | Mouse monoclonal anti-GFP | NeuroMab | N86-38 | 1:1000 |
| Antibody | rabbit polyclonal anti-LAMP1 | Novus | NB120-19294 | 1:1000 |
| Antibody | Alexa Fluor 488 polyclonal Goat anti-mouse | Life Technologies | A-11001 | 1:2000 |
| Antibody | Alexa Fluor 568 polyclonal Goat anti-rabbit | Life Technologies | A-11011 | 1:2000 |
| Recombinant DNA reagent | 293 fectin | Invitrogen | | |
| Recombinant DNA reagent | polyethyleneimine | SIGMA-Aldrich | | |
| Recombinant DNA reagent | Lipofectamine 3000 | Life Technologies | Cat #A10262 | |
| Chemical compound, drug | Ezetimibe | Santa Cruz Biotechnology | sc-205690 | |
| Software, algorithm | Flowjo | https://www.flowjo.com/ | | |
| Software, algorithm | FIJI | PMID:22743772, DOI: 10.1038/nmeth.2019 | | |

## Materials

Cholesterol, U18666a, sodium taurocholate, mono-olein, oleic acid, phosphatidyl-choline (2-oleoyl-1-palmityl-sn-glycero-3-phosphocholine) were from Sigma-Aldrich (St. Louis, MO); ezetimibe and lyso-phosphatidylcholine (1-palmitoyl-sn-glycero-3-phosphocholine) were from Santa Cruz Biotechnology (Santa Cruz, CA); Ni-NTA resin was from Qiagen (Valencia, CA), NHS-activated Sepharose 4 Fast Flow and Q-Sepharose Fast Flow was from GE Healthcare Life Sciences. $^3$H-cholesterol was from American Radiolabeled Chemicals (St. Louis, MO) and $^3$H-ezetimibe was from Merck. EZ-link Sulfo-NHS-SS-Biotin, SF900 III SFM insect cell medium, Freestyle 293 expression medium, Dulbecco's modified Eagle's medium (DMEM) and neutravidin agarose were from Life Technologies (Carlsbad, CA); lipoprotein-deficient serum was from KALEN Biomedical (Montgomery Village, Maryland). Chicken anti-GFP antibodies were from Life Technologies (used at 1:1000 for immunoblots). IRDye 680RD donkey anti-chicken and IRDye 800CW streptavidin were from LI-COR (Lincoln, NE) and used at 1:10,000 for immunoblots. Mouse anti-GFP antibody was from NeuroMab and Rabbit anti-LAMP1 antibody was obtained from Novus; both were used at 1:1000 for immunofluorescence. Alexa Fluor 488 Goat anti-mouse antibody and Alexa Fluor 568 Goat anti-rabbit antibody were obtained from Life Technologies and used at 1:2000 for immunofluorescence.

## Plasmids

cDNAs encoding full-length mouse NPC1-eGFP and human NPC1L1-eGFP and ΔN-NPC1L1 mutant (lacks residues 18–260) were described (*Johnson and Pfeffer, 2016*); P249K, P259K, P251C, L929C,

A521C, K1013C, Δ807–811, 807-811Ala mutants of NPC1 and A179K, L427N, N527D, N527K, K533E, E939K, K1026R, L1029N, L987N and R1002E mutants of NPC1L1 were generated by Quik-change mutagenesis. Rationale for the generation of NPC1L1 mutants is presented in Table 2. pFastBac NPC1L1-WT-N-terminal domain plasmid was from AddGene. The pCMV-FLAG-His6-NPC1L1-MLD was cloned in the pFLAG-CMV-3 plasmid (Sigma). This construct contains a pre-protrypsin signal sequence, a FLAG tag, and a His tag; NPC1L1 residues 373–632 of the human precursor were then flanked by sequences that form a stable, antiparallel coiled coil (*Deffieu and Pfeffer, 2011*).

## Cell culture

NPC1 knockout HeLa cells were generated using CRISPR to the target sequence AAAGAGTTACAA TACTACGT in exon 4 (*Li et al., 2017a*) and grown in DMEM with 7.5% (v/v) FBS. HEK293T cells (ATCC) were cultured in Dulbecco's modified Eagle's medium with 7.5% (v/v) FBS. HEK293F suspension cells were cultured in Freestyle 293 medium; all cells were grown at 37°C with 5% $CO_2$ except for SF9 insect cells which were cultured in SF900 III SFM medium at 27°C. Parental cells were obtained from ATCC to ensure identity and were checked routinely for mycoplasma using either MycoALert Mycoplasma Detection Kit (Lonza LT07-318) or PCR.

## Assay for cholesterol export from lysosomes

NPC1$^{-/-}$ cells were transfected with wild type or mutant NPC1 plasmids with lipofectamine 3000 (Life Technologies) according to the manufacturer. About 30 h post transfection, cells were incubated with 1 μM U18666a in 7.5% FBS-containing medium for ~16 h at 37°C to block cholesterol export from lysosomes. U18666a-containing medium was then removed and cells were washed once with phosphate buffered saline (PBS). U18666a export was enhanced using 5 min incubation in the presence of 10 mM methylamine hydrochloride (SIGMA) in DMEM medium with 7.5% FBS. Cells were then washed once with PBS to remove any residual serum and chased for 1 h in DMEM with 5% lipoprotein-deficient serum (LPDS) to permit cholesterol export from lysosomes. Lysosomal cholesterol was then immediately monitored using fluorescently labeled PFO (*Li et al., 2017a*) with immunofluorescence and/or flow-cytometry. In this assay, loss of PFO staining is not detected over the first 30 min of LPDS chase.

## Light microscopy

Cells were plated on collagen (Sigma-Aldrich)-coated cover slips and transfected with the indicated plasmids. Cells were fixed with 3.5% paraformaldehyde for 15 min, permeabilized for 5 min with 0.1% saponin, and blocked with 1% BSA in PBS. Cells were then incubated with mouse anti-GFP antibody and rabbit anti-LAMP1 antibody for 1 h at room temperature, followed by 1 h incubation with goat anti-mouse 488, goat anti-rabbit 568 and 10 ug/ml Alexa-647 labeled perfringolysin O (*Li et al., 2017a*). Coverslips were mounted using Mowiol with DAPI (Vector Laboratories) and imaged using a Leica SP2 confocal microscope and Leica software or an Olympus IX70 microscope with a 60 × 1.4 N.A. Plan Apochromat oil immersion lens (Olympus) and a charge-coupled device camera (CoolSNAP HQ, Photometrics). Images were analyzed using ImageJ, and nuclei were stained using 0.1 μg/ml DAPI (SIGMA).

## Flow cytometry

All steps were performed at room temperature. Cells were trypsinized, fixed, and labeled with PFO* as described above. Cells in PBS were then analyzed using a SONY SH800Z cell sorter (*Li et al., 2017a*).

## Membrane fractionation glycoform analysis

HEK293T cells were transfected with wild type and mutant NPC1 plasmids using polyethyleneimine (PEI). Forty-eight hours post transfection, cells were chilled on ice, washed with ice-cold PBS, and swelled in hypotonic buffer (10 mM HEPES pH 7.4). After 15 min, 5X buffer was added to achieve a final concentration of resuspension buffer (50 mM HEPES (pH 7.4), 150 mM NaCl, 1X protease inhibitor cocktail (Sigma)) and the suspension was passed 25 times through 25 gauge syringe. Nuclei were pelleted by centrifugation at 1,000Xg for 5 min at 4°C. The post-nuclear supernatant was

spun at 50,000Xg for 20 min in a tabletop ultracentrifuge in TLA100.2 rotor; the resulting supernatant was the cytosol fraction. Membrane pellets were solubilized in 0.5% Triton X100-containing 50 mM HEPES (pH 7.4), 150 mM NaCl, 1X protease inhibitor cocktail. Protein concentrations were estimated by Bradford assay (Bio-Rad, Richmond, CA). GFP-tagged NPC1 was captured on GFP binding protein (GBP) immobilized to NHS-activated sepharose for 2 h at room temperature. Immobilized proteins were deglycosylated with endoH in 0.5% SDS and 40 mM DTT (New England Biolabs) overnight at 37°C according to the manufacturer.

## LC-MS identification of NPC1 disulfide bonds

HEK293T cells were plated on 10 cm dishes. Twenty-four hours after plating, cells were transfected using polyethyleneimine with 6 µg of plasmid encoding indicated NPC1 constructs. Seventy-two hours post transfection, cells were washed and incubated for 4 h with 100 µg/mL cycloheximide, prior to membrane fractionation to enable turnover of any unfolded proteins in the endoplasmic reticulum. Membranes were permeabilized in lysis buffer (1% CHAPS, 50 mM HEPES pH 7.4, 150 mM NaCl). Samples were bound to immobilized GFP binding protein for 2 h at room temperature and washed 3X with 100 mM Tris-HCl pH 8.0, 2 mM $CaCl_2$. Samples were divided into two tubes and treated with 100 mM Tris-HCl pH 8.0, 2 mM $CaCl_2$, 3 M urea, with or without 5 mM DTT. Samples were then treated with 20 mM iodoacetamide in 100 mM Tris-HCl pH 8.0, 2 mM $CaCl_2$, 1 M urea, and deglycosylated overnight with PNGaseF. Samples were washed once, and proteolysis was conducted with 350 ng LysC, 350 ng trypsin, and 350 ng chymotrypsin (all sequencing grade) overnight at 37°C in 1 M urea-containing buffer. Peptides were brought to a final concentration of 5% formic acid, desalted with ZipTips (Millipore), dried by vacuum centrifugation, and resuspended in 20 µl 0.1% formic acid.

Synthetic peptides (Elim Biopharm) were prepared as external standards for the proteolytic NPC1 peptides containing the engineered cysteines P251C (CQPPPPPMK) and L929C (NAAECDTY). For reduced standards, peptides were suspended in 180 mM borate pH 8.6 and 90 mM NaCl, reduced with 1 mM TCEP for 5 min, and alkylated with 10 mM iodoacetamide at room temperature in the dark for 45 min, when formic acid was added to 2.5%. For disulfide standards, 50 nmol CQPPPPPMK was treated with 50 nmol 5,5'-dithiobis-(2-nitrobenzoic acid) in 50 mM ammonium bicarbonate pH 8 for 5 min at room temperature. Then 50 nmol NAAECDTY was added at room temperature for 2 h, then formic acid was added to 5%. Reduced and oxidized peptides were desalted by ZipTip (Millipore), dried by vacuum centrifugation, and resuspended in 100 µl 0.1% formic acid.

Both NPC1 and synthetic peptide samples were analyzed by LC-MS using a Fusion Lumos mass spectrometer (Thermo Fisher Scientific) equipped with a Dionex Ultimate 3000 LC-system (Thermo Fisher Scientific). Peptides were separated by capillary reverse phase chromatography on a 24-cm reversed phase column (100 µm inner diameter, packed in-house with ReproSil-Pur C18-AQ 3.0 m resin (Dr. Maisch GmbH)) in 0.1% (v/v) formic acid using a two-step linear gradient with 4–25% buffer B (0.1% (v/v) formic acid in acetonitrile) for 40 min followed by 25–40% buffer B for 5 min. The Fusion Lumos system (Tune 3.3) was used in top speed data-dependent mode with a duty cycle time of 3 s. Full MS scans were acquired in the Orbitrap mass analyzer with a resolution of 120,000 (FWHM) and m/z scan range of 400–1500. A targeted mass list including disulfides and carbamidomethylated cysteine peptides was used and dynamic exclusion was disabled. Precursor ions with charge state 2–7 (for NPC1 P251C/L929C/P249K/P259K) or 1–7 (for NPC1 A521C/K1013C) and intensity threshold above 50,000 were selected for fragmentation using higher-energy collisional dissociation (HCD) with quadrupole isolation, using an isolation window of 1.2 m/z and normalized collision energy of 30%. HCD fragments were analyzed in the Orbitrap mass analyzer with a resolution of 15,000 (FWHM). In the same run precursor ions with charge state 3–11 and intensity threshold above 50,000 were selected for fragmentation using electron transfer dissociation with 20% of supplemental collision energy (EThcD). Calibrated charge-dependent parameters were enabled. The AGC targets for full Fourier transform mass analyzers (FTMS) scans and FTMS2 scans were set to standard, and the maximum injection time for full FTMS scans was set to auto and for FTMS2 scans was set to dynamic. Extracted ion chromatograms were integrated using the Byos disulfide workflow (Protein Metrics) with a precursor mass tolerance of 10 ppm and a fragment mass tolerance of 0.02 Da, with proteolysis at lysine, arginine, tryptophan, tyrosine, or phenylalanine, allowing up to two missed cleavages. For NPC1 A521C/K1013C, proteolysis after leucine was also allowed. The following modifications

were permitted: carbamidomethylation or disulfide formation at cysteine, oxidation at methionine and tryptophan, and deamidation at asparagine.

## LC-MS quantification of NPC1 disulfide bonds

NPC1 samples were prepared as before but split into four aliquots after Nanotrap capture: the experimental tube, a heavy/heavy control, a light/light control, and a reduced control. All samples were incubated at room temperature in 100 mM Tris-HCl pH 8.0, 2 mM CaCl$_2$, and 3M urea. The reduced control was treated with 1 mM Tris 2-carboxyethyl phosphine hydrochloride (TCEP) during this period. The samples were labeled with 20 mM iodoacetamide—either iodoacetamide-$^{13}$C$_2$, 2-D$_2$ (98 atom % D, 99 atom % $^{13}$C, Sigma-Aldrich) (heavy) or iodoacetamide with no isotope labels (light). Samples were alkylated for 2 h at room temperature in the dark on a thermomixer and washed to remove iodoacetamide. Next, all samples were treated with 1 mM TCEP reduction buffer for 30 min at room temperature in a thermomixer. Samples were then labeled again with either light or heavy iodoacetamide for 2 h at room temperature in the dark on a thermomixer. Samples were then deglycosylated, proteolyzed, and desalted as before, followed by vacuum centrifugation and resuspension in 0.1% formic acid.

NPC1 disulfide quantification samples were analyzed by online capillary nanoLC-MS/MS. Samples were separated on an in-house made 20 cm reversed phase column (100 µm inner diameter, packed with ReproSil-Pur C18-AQ 3.0 µm resin (Dr. Maisch GmbH)) equipped with a laser-pulled nanoelectrospray emitter tip. Peptides were eluted at a flow rate of 400 nl/min using a two-step linear gradient including 2–25% buffer B in 23 min and 25–40% B in 12 min (buffer A: 0.2% formic acid and 5% DMSO in water; buffer B: 0.2% formic acid and 5% DMSO in acetonitrile) in a Dionex Ultimate 3000 LC-system (Thermo Scientific). Peptides were then analyzed using a LTQ Orbitrap Elite mass spectrometer (Thermo Scientific). The mass spectrometer was operated in a data-dependent mode using an inclusion list targeting disulfide and carbamidomethylated cysteine peptides with a 10 ppm tolerance. Full MS scans were acquired in the Orbitrap mass analyzer with a resolution of 60,000 and m/z scan range of 340–1600. The top 20 most abundant ions with intensity threshold above 500 counts and charge states 2 and above were selected for fragmentation using collision-induced dissociation (CID) with isolation window of 2 m/z, normalized collision energy of 35%, activation Q of 0.25, and activation time of 5 ms. The CID fragments were analyzed in the ion trap with rapid scan rate. Dynamic exclusion was enabled with repeat count of 2 in 30 s and exclusion duration of 20 s. The AGC target was set to 1,000,000 and 5000 for full FTMS scans and ITMSn (multi-stage ion trap MS) scans, respectively. The maximum injection time was set to 250 ms and 100 ms for full FTMS scans and ITMSn scans, respectively.

Extracted ion chromatograms were integrated for each isotopic peak corresponding to the carbamidomethylated peptides CQPPPPPMK and NAAECDTY that form the engineered P251C-K929C disulfide. For each peptide, the isotope distribution of the heavy/light sample was manually fit to a linear combination of the control distributions for the heavy/heavy and light/light samples weighted by intensity, resulting in a measurement of the fraction of the sample protected from heavy carbamidomethylation by the disulfide bond.

## Protein purification

pFastBac NPC1L1-WT-N-terminal domain was used to make virus for infection of Sf9 insect cells. Seventy-two hours post infection, Sf9 cultures were spun down to remove cells and ammonium sulfate added to the supernatant to achieve 70% saturation. The resulting precipitate was suspended in 50 mM HEPES, pH 7.4, 150 mM NaCl, and incubated with Ni-NTA resin overnight at 4°C. After elution with 50 mM HEPES, pH 7.4, 150 mM NaCl, 250 mM imidazole, the protein was further purified using Q-Sepharose.

A 100 ml culture (1 × 10$^6$ cells/ml) of 293 F cells was transfected with 100 µg of pCMV-FLAG-His6-NPC1L1-MLD plasmid using 293-fectin (Invitrogen) according to the manufacturer. The medium was collected 72 h after transfection and incubated overnight with Ni-NTA resin at 4°C. The protein was eluted in 50 mM HEPES, 150 mM NaCl, 250 mM imidazole.

## Cholesterol binding of NPC1L1 domains

80 µl solutions containing His-tagged NPC1L1 N-terminal domain or Domain2 (150 nM) were incubated with cholesterol (5 µM, 1:1000 $^3$H:$^1$H) in Buffer A (50 mM HEPES (pH 7.4), 150 mM NaCl), containing either 0.004% NP-40 and BSA (192 nM) or mixed bile salt micelles for 30 min at 37°C. Micelle lipids were first dissolved in ethanol, combined in a glass vial, and solvents evaporated under N$_2$. The dried lipids were incorporated into buffer A containing 24 mM sodium taurocholate by vortexing for 5 min. The solution was then diluted using buffer A to final concentrations: 5 mM sodium taurocholate, 0.5 mM oleic acid, 0.035 mM phosphatidylcholine, 0.08 mM lysophosphatidylcholine, 0.3 mM monoolein, 0.005 mM cholesterol, and 5 nM radiolabeled cholesterol (*Johnson and Pfeffer, 2016*). After incubation, solutions were loaded onto 1 ml syringes fitted with a frit and 20 µl Ni-NTA resin. After 15 min, the resin was washed 6X with 1 ml wash buffer (buffer A + 10 mM imidazole + 0.004% NP-40) and eluted with 1.2 ml elution buffer (buffer A + 250 mM imidazole + 0.004% NP-40). Eluted samples were mixed with BioSafe-II (Research Products International, Mt. Prospect, IL), and radioactivity was determined using a scintillation counter.

## Cell based $^3$H-ezetimibe binding assay

HEK293T cells plated on collagen-coated 12-well plates were transfected with wild type or mutant human NPC1L1-eGFP plasmids, three wells per condition. Forty-eight hours post transfection, cells were incubated with 50 nM $^3$H-ezetimibe under normal growth conditions for 2 h at 37°C. Non-specific binding was measured in the presence of 10 µM ezetimibe. At the end of the incubation, Ezetimibe was removed and cells were washed two times with 500 µl ice cold 1 mM MgCl$_2$ + 0.1 mM CaCl$_2$ in PBS (pH 7.4). Cells were lyzed with 100 µl, 0.5% TritonX-100, 50 mM HEPES (pH 7.4), 150 mM NaCl, 1X protease inhibitor cocktail. Cleared lysates were assayed for total protein and equal lysate protein amounts were analyzed for bound $^3$H-ezetimibe using a scintillation counter.

## Cell surface biotinylation and cholesterol uptake

The fraction of total NPC1L1 present on the cell surface was measured as described (*Weixel and Bradbury, 2002*): HEK293T cells were plated on 6-cm collagen-coated dishes. Twenty-four hours after plating, cells were transfected with 3 µg wild type or mutant NPC1L1 plasmids using PEI; 48 h post transfection, media was removed and cells were washed 3X with ice cold PBS on ice. Cell surface proteins were labeled with 1 mg/ml sulfo-NHS-biotin-EZ link for 30 min. Excess biotin reagent was removed by washing 3X with 1% BSA in PBS, pH 7.4 on ice. Cells were lysed with buffer containing 0.5% Triton-X 100, 50 mM HEPES (pH 7.4), 150 mM NaCl and 1X protease inhibitor cocktail. Cleared lysates were incubated overnight with 25 µl NeutrAvidin resin, then spun at 700X*g* for 3 min and loaded into a 1 ml syringe fitted with a frit. Resin was washed 5X with cold wash buffer (0.5% Triton-X 100, 50 mM HEPES (pH 7.4), 150 mM NaCl). Syringes were spun at 1000 rpm for 5 min to remove wash buffer; the resin was then transferred into a 1.5-ml tube containing 100 µl SDS–PAGE sample buffer and heated for 20 min at 37°C. Duplicate 40-µl portions of each sample were analyzed on Bio-Rad (Hercules, CA) Mini-PROTEAN TGX 4–20% gradient gels. After transfer to nitrocellulose membrane and antibody incubation, immunoblots were visualized using a LICOR and analyzed using ImageJ software (National Institutes of Health, Bethesda, MD). Cholesterol uptake was monitored as described (*Johnson and Pfeffer, 2016*).

## Model construction

The wild type NPC1 protein structure was modeled as follows. The protein atomic coordinates were constructed from two sets of data. Residues 23–288 were taken from the lower resolution (4.43 Å) cryo-EM PDB structure 3jd8 (*Gong et al., 2016*) and residues 334–1255 were taken from PDB structure 5u74 (3.33 Å; *Li et al., 2017b*). The two structures were overlapped using Chimera (*Pettersen et al., 2004*) and the new protein coordinates were saved. Missing internal residues (289-333, 642-649, 800-813) were reconstructed using a combination of CHARMM (*Brooks et al., 1983*) and the CHARMM-GUI (*Jo et al., 2008*). Hydrogen atoms were added using H-build from CHARMM and the N- and C-termini were capped with neutral groups CH3–CO– and methyl acetate –NH–CO–OCH3, respectively. Based on the crystal structures, we constructed 15 disulfide bonds in WT NPC1 using CHARMM, [Cys25-Cys74, Cys31-Cys42, Cys63-Cys109, Cys75-Cys113, Cys97-Cys238, Cys100-Cys160, Cys177-Cys184, Cys227-Cys243, Cys240-Cys247, Cys468-Cys479,Cys516-Cys533, Cys909-

Cys914, Cys956-Cys1011, Cys957-Cys979, Cys967-Cys976]. In addition to the WT, additional mutant NPC1 structures were designed. These were constructed analogously to the WT, and the mutations were introduced using CHARMM. For the mutants P251C/L929C and A521C/K1013C, disulfide bonds were constructed between the mutated amino acids.

For the WT and mutant structures, we determined an initial protonation pattern by calculating the pKa values of all titratable residues. For this, electrostatic energy computations were carried out with karlsberg+ (*Kieseritzky and Knapp, 2008*). Based on these results, a model was built to represent the protonation pattern at pH 5. This model included the following non-standard protonated amino acid side-chains: His215, His441, His492, His510, His512, His758, His1016, His1029, His1170, Glu406, Glu688, and Glu742. Next, each protein structure was modeled in the lipid bilayer using the CHARMM-GUI and OPM database (*Lomize et al., 2006*). For this, a lipid bilayer consisting of cholesterol (10%), DOPG (dioleoylphosphatidylglycerol 10%), and POPC (1-palmitoyl-2-oleoyl-glycero-3-phosphocholine 80%) was constructed. The total solvated system, including $Na^+$ (287) and $Cl^-$ (194) ions to neutralize charge, and 99,785 explicit TIP3 water molecules (*Jorgensen et al., 1983*), had a total size of 399,269 atoms and was simulated in a rectangular box of dimension 143.5 Å x 143.5 Å x 184.9 Å.

## Geometry optimizations and molecular dynamics

The initial geometry of each solvated NPC1-membrane complex was optimized with 1500 steps of steepest descent (SD) energy minimization, followed by 1500 adopted basis Newton-Raphson (ABNR) (*Brooks et al., 1983*) steps to remove any close contacts. All energy minimizations and geometry optimizations used the all-atom CHARMM36 parameter set for the protein (*MacKerell et al., 1998*) and the TIP3P model for water molecules (*Jorgensen et al., 1983*).

The solvated protein-membrane complex was simulated with molecular dynamics (MD) at 310 K according to the following protocol: 1) equilibration MD with Langevin dynamics (time step of 1 fs) for 50 ps followed by CPT dynamics (time step 2 fs) for 350 ps; 2) production MD with CPT dynamics (time step 2 fs) for 30 ns. To simulate a continuous system, periodic boundary conditions were applied. Electrostatic interactions were summed with the Particle Mesh Ewald method (*Essmann et al., 1995*; grid spacing ~1.4 Å; fftx 150, ffty 150, fftz 192). A nonbonded cutoff of 16.0 Å was used, and Heuristic testing was performed at each energy call to evaluate whether the nonbonded pair list should be updated.

Triplicate MD runs (minimum 20 ns each) were carried out for each of the models. Statistical analysis was performed on each simulated trajectory, consisting of 20 ns of sampling, as well as on the statistical ensemble obtained averaging multiple trajectories (i.e., three replicas for each model). Each simulated trajectory was obtained from an independent MD simulation initiated from a unique set of atomic velocities, as initialized according to the Maxwell-Boltzmann distribution at physiological temperature.

## Calculation of distance correlation coefficients

Distance correlation coefficients (DiCC) were calculated using the dcor function in CHARMM (*Roy and Post, 2012*; *Brooks et al., 1983*). DiCC, calculated from distance covariance, have been shown to best capture the correlation between positional vectors and to be a valid measure of concerted atomic motions as they are least sensitive to angular dependence (*Roy and Post, 2012*). For two vector series {A} and {B} containing the atomic position from an MD trajectory, the DiCC between the two vectors is defined as

$$DiCC = \frac{v(A,B)}{\sqrt{v(A,A)v(B,B)}} \tag{1}$$

where v(A,B) is the distance covariance between the vectors and is defined as

$$v(A,B) = \sqrt{\frac{1}{n^2} \sum ij\, a_{ij}\, b_{ij}}. \tag{2}$$

The elements $a_{ij} = a_{ij} - a_{i\cdot} - a_{\cdot j} + a_{\cdot\cdot}$ are the elements of vector A, and $b_{ij}$ are analogously the elements of vector B. Using this approach, the motion of protein domains can be analyzed to evaluate the degree of long-range concerted motion, particularly for multi-domain proteins (*Roy et al.,*

*2016*). Here, the DiCC for each model were calculated from the MD trajectories by aligning first the protein Cα atoms and calculating the distance covariance for each pair of domains, based on the positions of Cα atoms.

## Other methods

Structure models were created using Chimera software (*Pettersen et al., 2004*); statistical analyses were carried out using PRISM 8 software.

## Acknowledgements

This research was supported by grants from the Ara Parseghian Medical Research Foundation and the NHLBI (5R01HL13499103) to SRP. Access to mass spectrometry analysis software was supported in part by NIH P30 CA124435 utilizing the Stanford Cancer Institute Proteomics/Mass Spectrometry Shared Resource. We thank Dr. Bharat Adkar for help identifying interface residues in NPC1L1 and Dr. Paulina Wawro for help with image presentation.

## Additional information

### Competing interests

Suzanne R Pfeffer: Reviewing editor, *eLife*. Piyali Saha: is affiliated with Merck Research Laboratories. The author has no financial interests to declare. This work was completed prior to her employment at Merck and this basic study will not impact any sales of Zetia, the tradename of Ezetimibe. Niclas E Olsson: is affiliated with Calico Life Sciences LLC. The author has no financial interests to declare. The other authors declare that no competing interests exist.

### Funding

| Funder | Grant reference number | Author |
|---|---|---|
| Ara Parseghian Medical Research Foundation | | Suzanne R Pfeffer |
| National Institutes of Health | 5R01HL134991-04 | Suzanne R Pfeffer |
| Chan Zuckerberg BioHub | | Joshua E Elias |

The funders had no role in study design, data collection and interpretation, or the decision to submit the work for publication.

### Author contributions

Piyali Saha, Conceptualization, Data curation, Formal analysis, Investigation, Visualization, Methodology, Writing - original draft, Project administration, Writing - review and editing; Justin L Shumate, Conceptualization, Data curation, Formal analysis, Validation, Investigation, Visualization, Writing - review and editing; Jenna G Caldwell, Conceptualization, Data curation, Formal analysis, Validation, Investigation, Visualization, Writing - original draft, Writing - review and editing; Nadia Elghobashi-Meinhardt, Conceptualization, Data curation, Formal analysis, Investigation, Visualization, Writing - original draft, Writing - review and editing; Albert Lu, Data curation, Formal analysis, Investigation, Visualization, Writing - review and editing; Lichao Zhang, Conceptualization, Data curation, Formal analysis, Validation, Investigation, Writing - review and editing; Niclas E Olsson, Resources, Data curation, Formal analysis; Joshua E Elias, Conceptualization, Resources, Formal analysis, Supervision; Suzanne R Pfeffer, Conceptualization, Data curation, Formal analysis, Supervision, Funding acquisition, Investigation, Visualization, Writing - original draft, Project administration, Writing - review and editing

### Author ORCIDs

Jenna G Caldwell  https://orcid.org/0000-0001-6392-4794
Nadia Elghobashi-Meinhardt  https://orcid.org/0000-0002-1023-6856

Lichao Zhang [ID] https://orcid.org/0000-0003-1811-7568
Niclas E Olsson [ID] https://orcid.org/0000-0002-9746-0542
Suzanne R Pfeffer [ID] https://orcid.org/0000-0002-6462-984X

**Decision letter and Author response**
Decision letter https://doi.org/10.7554/eLife.57089.sa1
Author response https://doi.org/10.7554/eLife.57089.sa2

## Additional files

### Supplementary files
• Transparent reporting form

### Data availability
All data generated or analysed during this study are included in the manuscript and supporting files.

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
