## [Decision Letter]

**Acceptance summary:**

This study provides important insights into how two large polytopic membrane proteins, NPC1 and NPC1L1, move cholesterol across the lipid bilayer of the plasma membrane and the lysosome, respectively. Using a combination of clever biochemistry and cell biology approaches, the authors demonstrate how concerted movements of various domains of NPC1 and NPC1L1 allow for the movement of cholesterol through the cores of the two proteins.

**Decision letter after peer review:**

[Editors’ note: the authors submitted for reconsideration following the decision after peer review. What follows is the decision letter after the first round of review.]

Thank you for submitting your work entitled "Inter-domain dynamics drive cholesterol transport by NPC1 and NPC1L1 proteins" for consideration by *eLife*. Your article has been reviewed by a Senior Editor, a Reviewing Editor, and three reviewers. The reviewers have opted to remain anonymous.

Our decision has been reached after consultation between the reviewers. Based on these discussions and the individual reviews below, we regret to inform you that your work will not be considered further for publication in *eLife*.

Your study examines the important problem of how NPC1 and NPC1L1 proteins move cholesterol across membranes. Using knowledge gleaned from prior structural and biochemical studies of NPC1 and NPC1L1, you have carried out interesting protein engineering studies to show how movements of domains of either protein facilitate movement of insoluble cholesterol through the protein. While the reviewers all agreed that the study was interesting, they raised several concerns, addressing which would take more than the two month revision window. The reviewers welcome assessment of a revised submission, but only if it addressed all the points raised in the full reviews listed below.

Reviewer #1:

The manuscript by Saha et al. reports that multiple domain dynamics drive cholesterol transport by NPC1 and NPC1L1. The authors introduced disulfide bonds into NPC1 to block inter-domain dynamics, then performed PFO* staining to detect cholesterol accumulation in lysosomes after transfecting the mutant plasmids into NPC1-KO HeLa cells. Also, the authors performed molecular dynamics simulations to further support the model that cholesterol can pass directly through NPC1. In addition, the authors showed that the NTD and MLD of NPC1L1 can bind cholesterol in different environments. It appears very interesting and provides potential insights into NPC1L1-mediated dietary cholesterol uptake. However, the interpretation and analysis of the experiments presented here is questionable and additional experiments are required to support the conclusions.

Essential revisions:

1) The membrane fractionation glycoform analysis appears too weak to demonstrate that a disulfide bridge had formed between P251C and L929C. The amount of under-glycosylated NPC1 that is shown after endo H treatment does not mean that C251 and C929 have formed a disulfide bridge. It is possible that L929C affects the conformation of the domain 3 alone, which includes the most glycosylation sites of NPC1. The authors should use proteolysis and mass spec to verify the disulfide bridge.

2) The construct A521C/ K1013C also requires proteolysis and mass spec to identify formation of the disulfide bridge.

3) In Figure 1E, the PFO* fluorescence curve between NPC1^-/-^ and wild type is significantly separate; while in Figure 3D, the difference is not as evident. It seems that the PFO* fluorescence assay is not stable or repeatable for this analysis. Can the author use more quantitative methods or modify their assay to make it clearer?

4) There is no data to show protein expression levels. It is possible that the mutants cannot transport cholesterol well due to the reduced protein expression rather than the domain locking.

5) The authors claimed that domain 2 of NPC1L1 can function to bind bile salt micelles; in contrast, the NTD of NPC1L1 does not. It is quite an interesting and novel observation. However, to further support this point, additional control assays are required. The authors may consider using some other 3H-steroid or 3H-25-hydroxycholesterol in NP40 and in bile salt micelles to repeat the binding assay. If domain 2 can still bind to those unrelated molecules, domain 2 may only recognize bile salt micelles; if not, cholesterol may also contribute to bind domain 2.*Reviewer #2:*This paper examines the mechanism by which the NPC1 protein moves cholesterol from a region outside the glycocalyx toward or into the limiting membrane. The paper takes advantage of prior structural studies and studies of cholesterol binding sites in NPC1. The authors interpret their data as being consistent with a model in which movements of domains of the protein facilitate cholesterol movement though the protein. They also suggest that a similar mechanism may operate for NPC1L1, a protein on intestinal cells that imports dietary cholesterol.

While the experiments are in general well described and provide interesting data, this reviewer is not convinced that the main conclusions of the paper are justified by the data.

1) The NPC1 protein with deletion Δ807-811 is ineffective at removing cholesterol from lysosomes. This is interpreted as requiring flexibility in the loop between TM7 and TM8. However, a change in the loop length could stably alter the interaction between these transmembrane domains either by changing the angle between them or by changing the penetration of one or both into the bilayer.

2) Similarly, crosslinking domains 2 and 3 might simply lock the protein into a conformation that does not allow cholesterol binding or movement. This does not show that there is a hand-off or similar concerted motion as occurs with transporters as described in the Introduction. It may just be that protein motion is required to expose cholesterol binding sites.

3) For NPC1L1 it is claimed that experiments with shortening the loop shows the importance of interdomain flexibility. The issues noted in comment 1 apply here also. Additionally, the distance correlation coefficients were not greatly altered in the NOC1 simulation with the deletion, and the RMSD vales were actually increased in the deletion mutant.

4) The experiments with domains of NPC1L1 are not conclusive. First, it may be that binding of cholesterol from bile salt micelles to Domain 2 may be very different in the presence or absence of the NTD and Domain 3.

5) The experiment on the effect of deleting the NTD on Ezetimibe binding does indicate that more than Domain2 is required for binding this inhibitor.

Overall, this paper has several interesting observations, but this reviewer does not feel that they justify the conclusions drawn about how NPC1 moves cholesterol.

*Reviewer #3:*This is a very interesting paper which uses a number of different approaches, including several clever biochemical experiments, to provide novel insights into the mechanisms of action of two key proteins involved in cellular cholesterol transport, NPC1 and NPC1L1. Introduction of several disulfide bonds to restrict domain movements was used to show that the N-terminal domain of NPC1 does not, as had been proposed previously, undergo a large conformational movement to transfer its bound cholesterol to the sterol sensing domain on either itself or on an adjacent NPC1 protein; rather, the results indicate that conformational changes in domains 2 and 3 must occur to allow normal cholesterol efflux to occur. A further key finding is that a short flexible loop on the cytoplasmic side of the protein is also required for sterol transport; these two observations imply that cholesterol is actually transiting through a presumably transient pore in the NPC1 structure. For NPC1L1 several similar experiments (namely introducing disulfides to constrain domains 2 and 3, and mutagenesis of the cytoplasmic loop) indicate that its inhibitor Ezetimibe binds to several domains and may thus act as a 'stopper' to prevent cholesterol from entering/generating a transmembrane pore.

This paper will undoubtedly be of high interest to researchers involved in studying cellular cholesterol homeostasis. There are nevertheless a number of comments that could be addressed to provide further support for some of the interpretations made and to clarify certain aspects of the presentation.

1) The kinetic assay for cholesterol trafficking which uses the recovery from U18666A-mediated cholesterol accumulation in NPC1 null cells transfected with WT and various mutants is said to be on a 'short' time scale; from the results included this seems to be about an hour. Recovery (disappearance of cholesterol staining) is used to conclude that a particular construct works or doesn't work. The results would be strengthened by (a) including a true time course and perhaps establishing rates of cholesterol mobilization, rather than a single time point for evaluation – this may allow additional insight into the various mutant proteins, e.g. showing which aspect of NPC1 or NPC1L1 motion might be of primary importance; and (b) demonstrating what happens over much longer time courses – is the NTD-polylinker-domain 3 mutant truly not at all different than the WT protein, or is there some possibility that the NTD conformation may be involved in the transport process? Overall, a snapshot at a single point in time is not really robust enough to draw 'kinetic' conclusions, as presented.

2) A very important aspect for interpretation of the data for the disulfide mutants is demonstration that these bridges had actually formed, as the authors acknowledge (otherwise the fact that they function like WT protein may simply be because they haven't formed the disulfide). The use of changes in glycosylation patterns is indirect but supportive evidence. Short of actually expressing and purifying the proteins, which would be ideal but quite challenging, the authors may wish to refer to their DiCC results presented in subsection “Molecular Dynamics Simulations of NPC1 and mutant proteins”, where they show that expected changes in interdomain mobility seem to occur.

3) NPC1L1 studies. It would be of interest to include a linker-domain 3 pair, similar to what was done for NPC1 – is the movement of the N-terminal domain also not important in NPC1L1?

4) Please add the control single cysteine mutant flow cytometry data to Figure 3D, presumably it appears like WT.

5) Point mutagenesis of NPC1L1 (Figure 8) shows numerous residues are essential for Ezetimibe binding. The rationale for the mutations chosen is not presented – why were some residues mutated to K, others to N, to D, to E, and to R? And, other than cell surface localization of the entire protein, is there a way of determining that the domains mutated are still folding properly?

6) In several instances the description of the figures in the text are a bit difficult to follow and the figure legends could be more specific:

a) Figure 1. Results section denotes the putative sterol sensing domain – this is not highlighted on the figure but should be. Also, it would be helpful if Figure 1B also used the color scheme to identify the domains, and also indicated the plane of the bilayer.

b) Figure 1 legend should include the time point following U18666A washout that the images were taken at, and define what "briefly incubated" with TMA-HCl means. Results section, please specify what is meant by "a much shorter time scale."

c) Same for Figure 2, at what time point following U-washout were images taken?

d) Molecular dynamics figure (Figure 4) and its related text could be more congruent.

7) For molecular dynamics, please include how many times was each experimental trajectory run?

8) Why was Nonidet P40 chosen as a control for the mixed micelles rather than the micelles below their CMC?

[Editors’ note: further revisions were suggested prior to acceptance, as described below.]

Thank you for re-submitting your article "Inter-domain dynamics drive cholesterol transport by NPC1 and NPC1L1 proteins" for consideration by *eLife*. Your article has been reviewed by Vivek Malhotra as the Senior Editor, a Reviewing Editor, and three reviewers. The reviewers have opted to remain anonymous.

The reviewers have discussed the reviews with one another and the Reviewing Editor has drafted this decision to help you prepare a revised submission.

We would like to draw your attention to changes in our revision policy that we have made in response to COVID-19 (https://elifesciences.org/articles/57162). Specifically, we are asking editors to accept without delay manuscripts, like yours, that they judge can stand as *eLife* papers without additional data, even if they feel that they would make the manuscript stronger. Thus, the revisions requested below only address clarity and presentation.

Summary:

In this revised version, Saha et al. have addressed all the concerns raised in the review of the first submission. All reviewers were enthusiastic about this paper. They all agreed that this revised manuscript was strong and provided new insights into how cholesterol can be transported through the molecular channel of NPC1 and NPC1L1. A couple of concerns were raised that the authors should be able to address with clarifying explanations and/or additional data already in hand.

Essential revisions:

1) Please rephrase the interpretation of the Δ807-811 mutant. The deletion of this loop could be due to altered mobility of the TM helices (as the authors propose), however the data do not rule out the possibility that this deletion favors a conformation of the protein that does not facilitate cholesterol transport. Distinguishing these two possibilities (which may be intertwined) is difficult and no new experiments are suggested, merely a rewording of the interpretation.

2) The images showing colocalization of NPC1 with LAMP1 are not convincing. The images are almost all dominated by a large unresolved object near to or surrounding the nucleus. This is a region that is dense with other organelles, so it is not clear whether two proteins in this region are indeed in the same organelles. There are some peripheral dots, but with the brightness adjusted for the large object they are very hard to see. We suggest providing insets of higher magnification and with brightness adjusted so that one can see clearly resolved lysosomes. This is not as important for the mutants that work to remove cholesterol, but is important for the mutants that are defective.

---

## [Author Response]

[Editors’ note: the authors resubmitted a revised version of the paper for consideration. What follows is the authors’ response to the first round of review.]

Reviewer #1:Essential revisions:1) The membrane fractionation glycoform analysis appears too weak to demonstrate that a disulfide bridge had formed between P251C and L929C. The amount of under-glycosylated NPC1 that is shown after EndoH treatment does not mean that C251 and C929 have formed a disulfide bridge. It is possible that L929C affects the conformation of the domain 3 alone, which includes the most glycosylation sites of NPC1. The authors should use proteolysis and mass spec to verify the disulfide bridge.

We have now added two types of mass spec data to document the presence of the disulfide bond. This took ~18 months but the final data are very clean (new Figure 2, Figure 2—figure supplement 1, and Figure 4—figure supplement 1) and we agree that this makes our case much stronger. We needed to introduce a few amino acid changes to facilitate proteolysis for mass spec peptide recovery; we demonstrate that the changed protein is fully functional, properly localized and contains a clearly documented disulfide bond.

2) The construct A521C/ K1013C also requires proteolysis and mass spec to identify formation of the disulfide bridge.

We now show mass spec data for one of the peptides under reducing and non-reducing conditions, consistent with 85% disulfide bond formation. We have also added the flow cytometry of the single mutant as requested.

3) In Figure 1E, the PFO* fluorescence curve between NPC1-/- and wild type is significantly separate; while in Figure 3D, the difference is not as evident. It seems that the PFO* fluorescence assay is not stable or repeatable for this analysis. Can the author use more quantitative methods or modify their assay to make it clearer?

The method is highly quantitative as we compare directly, only similar GFP expression levels for each construct (gating the fluorescence per cell), for *hundreds* of cells expressing similar levels (and clarified the text). We have replaced Figure 3D with a better example. The challenge is that the NPC1 KO cells are hard to transfect due to their accumulation of cholesterol in the endocytic pathway.

4) There is no data to show protein expression levels. It is possible that the mutants cannot transport cholesterol well due to the reduced protein expression rather than the domain locking.

We apologize that this has not been more clearly described – we gate the flow cytometry analysis according to construct expression of all constructs so that they are compared at equal expression levels. We have clarified the text and legends to make this more clear.

5) The authors claimed that domain 2 of NPC1L1 can function to bind bile salt micelles; in contrast, the NTD of NPC1L1 does not. It is quite an interesting and novel observation. However, to further support this point, additional control assays are required. The authors may consider using some other 3H-steroid or 3H-25-hydroxycholesterol in NP40 and in bile salt micelles to repeat the binding assay. If domain 2 can still bind to those unrelated molecules, domain 2 may only recognize bile salt micelles; if not, cholesterol may also contribute to bind domain 2.

We only conclude that domain 2 (MLD) can bind cholesterol when presented in the form of a bile salt micelle.

Additional binding experiments are beyond the scope of this story and would require significant additional work.

Reviewer #2:1) The NPC1 protein with deletion Δ807-811 is ineffective at removing cholesterol from lysosomes. This is interpreted as requiring flexibility in the loop between TM7 and TM8. However, a change in the loop length could stably alter the interaction between these transmembrane domains either by changing the angle between them or by changing the penetration of one or both into the bilayer.

Yes of course but we designed it based on the spacing expected from the crystal structure to try to avoid any strain and we have clarified the text accordingly. This is a small piece of the total 15 residue unstructured domain. It is striking that replacement of these residues with alanine of the same length was fully functional, further suggesting that no protein recognition on the cytoplasmic face is needed there for function. We tried to clarify the text so that there is no misunderstanding.

2) Similarly, crosslinking domains 2 and 3 might simply lock the protein into a conformation that does not allow cholesterol binding or movement. This does not show that there is a hand-off or similar concerted motion as occurs with transporters as described in the Introduction. It may just be that protein motion is required to expose cholesterol binding sites.

Agreed and we have clarified the text – we only meant to conclude that movement of some sort is needed.

3) For NPC1L1 it is claimed that experiments with shortening the loop shows the importance of interdomain flexibility. The issues noted in comment 1 apply here also. Additionally, the distance correlation coefficients were not greatly altered in the NOC1 simulation with the deletion, and the RMSD vales were actually increased in the deletion mutant.

Regarding the simulation, any mutant that constrains the protein at the bottom might be expected to yield increased mobility elsewhere. However, the RMSD increase was not that much more than that seen for the protein constrained at the top. We are not trying to conclude too much from the simulations; the mutant data were nevertheless surprising.

4) The experiments with domains of NPC1L1 are not conclusive. First, it may be that binding of cholesterol from bile salt micelles to Domain 2 may be very different in the presence or absence of the NTD and Domain 3.

Sure. But please note that no one has previously shown bile salt micelle binding to any domain of NPC1L1. Our work on domain 2 (MLD) of NPC1 has completely been recapitulated with the full length protein in subsequent experiments which motivated these experiments.

5) The experiment on the effect of deleting the NTD on Ezetimibe binding does indicate that more than Domain2 is required for binding this inhibitor.

Yes we agree.

Reviewer #3:1) The kinetic assay for cholesterol trafficking which uses the recovery from U18666A-mediated cholesterol accumulation in NPC1 null cells transfected with WT and various mutants is said to be on a 'short' time scale; from the results included this seems to be about an hour. Recovery (disappearance of cholesterol staining) is used to conclude that a particular construct works or doesn't work. The results would be strengthened by (a) including a true time course and perhaps establishing rates of cholesterol mobilization, rather than a single time point for evaluation – this may allow additional insight into the various mutant proteins, e.g. showing which aspect of NPC1 or NPC1L1 motion might be of primary importance; and (b) demonstrating what happens over much longer time courses – is the NTD-polylinker-domain 3 mutant truly not at all different than the WT protein, or is there some possibility that the NTD conformation may be involved in the transport process? Overall, a snapshot at a single point in time is not really robust enough to draw 'kinetic' conclusions, as presented.

This would be great but the experiments are harder because the cells are difficult to transfect efficiently and we are now locked out of our lab for at least the next month due to COVID-19. We have softened the text to downplay the assay but our controls are all here and signals are all wild type protein dependent – no difference is seen at 30 minutes and assays are all at 60 minutes.

2) A very important aspect for interpretation of the data for the disulfide mutants is demonstration that these bridges had actually formed, as the authors acknowledge (otherwise the fact that they function like WT protein may simply be because they haven't formed the disulfide). The use of changes in glycosylation patterns is indirect but supportive evidence. Short of actually expressing and purifying the proteins, which would be ideal but quite challenging, the authors may wish to refer to their DiCC results presented in subsection “Molecular Dynamics Simulations of NPC1 and mutant proteins”, where they show that expected changes in interdomain mobility seem to occur.

As mentioned above, we have carried out mass spec to now demonstrate the disulfide bonds.

3) NPC1L1 studies. It would be of interest to include a linker-domain 3 pair, similar to what was done for NPC1 – is the movement of the N-terminal domain also not important in NPC1L1?

Yes this would be of interest but it is beyond the scope of the present story. Each mutant requires proving that the protein is correctly localized and that the disulfide has actually formed which is a huge amount of work.

4) Please add the control single cysteine mutant flow cytometry data to Figure 3D, presumably it appears like WT.

Added as requested.

5) Point mutagenesis of NPC1L1 (Figure 8) shows numerous residues are essential for Ezetimibe binding. The rationale for the mutations chosen is not presented – why were some residues mutated to K, others to N, to D, to E, and to R? And other than cell surface localization of the entire protein is there a way of determining that the domains mutated are still folding properly?

We have now included detailed rationale for the mutagenesis (Table 2). Transit through the secretory pathway is perhaps the most stringent test of proper protein folding.

6) In several instances the description of the figures in the text are a bit difficult to follow and the figure legends could be more specific:a) Figure 1. Results section denotes the putative sterol sensing domain – this is not highlighted on the figure but should be.

Done.

Also, in the figure, it would be helpful if Figure 1B also used the color scheme to identify the domains, and also indicated the plane of the bilayer.

Done.

b) Figure 1 legend should include the time point following U18666A washout that the images were taken at, and define what "briefly incubated" with TMA-HCl means. Results section, please specify what is meant by "a much shorter time scale."c) Same for Figure 2, at what time point following U-washout were images taken?

Done.

d) Molecular dynamics figure (Figure 4) and its related text could be more congruent.

Done.

7) For molecular dynamics, please include how many times was each experimental trajectory run?

Now included.

8) Why was Nonidet P40 chosen as a control for the mixed micelles rather than the micelles below their CMC?

We wanted to show that cholesterol alone did not bind whereas when it is presented with other bile salts, it does bind. The sub-CMC NP40-cholesterol binds to NPC1 NTD and NPC2; bile salts don't. We have clarified the text.

[Editors’ note: what follows is the authors’ response to the second round of review.]

Essential revisions:1) Please rephrase the interpretation of the Δ807-811 mutant. The deletion of this loop could be due to altered mobility of the TM helices (as the authors propose), however the data do not rule out the possibility that this deletion favors a conformation of the protein that does not facilitate cholesterol transport. Distinguishing these two possibilities (which may be intertwined) is difficult and no new experiments are suggested, merely a rewording of the interpretation.2) The images showing colocalization of NPC1 with LAMP1 are not convincing. The images are almost all dominated by a large unresolved object near to or surrounding the nucleus. This is a region that is dense with other organelles, so it is not clear whether two proteins in this region are indeed in the same organelles. There are some peripheral dots, but with the brightness adjusted for the large object they are very hard to see. We suggest providing insets of higher magnification and with brightness adjusted so that one can see clearly resolved lysosomes. This is not as important for the mutants that work to remove cholesterol, but is important for the mutants that are defective.

We are grateful for the support of the reviewers, we have done everything asked and hope this story is now suitable for publication in *eLife*. Figure 3B and Figure 4B now include enlarged insets as requested, to make co-localization easier to see, it was a great suggestion.